# Neutron scattering and neural-network quantum molecular dynamics investigation of the vibrations of ammonia along the solid-to-liquid transition

T. M. Linker[1,2], A. Krishnamoorthy[3], L. L. Daemen[4], A. J. Ramirez-Cuesta [4], K. Nomura[1], A. Nakano [1], Y. Q. Cheng [4] ✉, W. R. Hicks[4], A. I. Kolesnikov [4] ✉ & P. D. Vashishta [1] ✉

Vibrational spectroscopy allows us to understand complex physical and chemical interactions of molecular crystals and liquids such as ammonia, which has recently emerged as a strong hydrogen fuel candidate to support a sustainable society. We report inelastic neutron scattering measurement of vibrational properties of ammonia along the solid-to-liquid phase transition with high enough resolution for direct comparisons to ab-initio simulations. Theoretical analysis reveals the essential role of nuclear quantum effects (NQEs) for correctly describing the intermolecular spectrum as well as high energy intramolecular N-H stretching modes. This is achieved by training neural network models using ab-initio path-integral molecular dynamics (PIMD) simulations, thereby encompassing large spatiotemporal trajectories required to resolve low energy dynamics while retaining NQEs. Our results not only establish the role of NQEs in ammonia but also provide general computational frameworks to study complex molecular systems with NQEs.

Properties of molecular crystals and liquids such as water and ammonia are dictated by complex dynamics of hydrogen bonding, dispersion interactions, and many-body polarization effects whose understanding remain a fundamental scientific problem[1–7]. Ammonia (NH₃) has garnered much attention due not only to its rich "water-like" nature but also due to a wide range of applications such as room-temperature synthesis of metal chalcogenides and development of exotic liquid metals[8,9]. In particular, over the past few years, there has been a growing movement for the development of green ammonia-based fuel technologies[10–12]. Ammonia has a higher energy density, at 12.7 MJ/L, than even liquid hydrogen, at 8.5 MJ/L. Liquid hydrogen must be stored at cryogenic conditions of −253 °C, whereas ammonia can be stored at a much less energy-intensive −33 °C (at $P = 25$ bar).

Furthermore, thanks to a century of ammonia use in agriculture, a vast ammonia infrastructure already exists. Worldwide, some 180 million metric tons of ammonia are produced annually, and 120 ports are equipped with ammonia terminals. The development of technologies based on ammonia will be reliant on our ability to understand and model the complex physical and chemical interactions that give rise to its unique properties. However, much is still not understood about the dynamics of ammonia. For example, whether ammonia hydrogen bonds in its solid and liquid phases is still a matter of open debate[2–7].

Vibrational spectroscopy often offers a window into signatures of molecular crystals and liquid's physical and chemical interactions[6,13,14]. The collective vibrational behavior of an ensemble of molecules (e.g., in clusters, solids, and liquids) is critical for predicting their

[1]Collaboratory for Advanced Computing and Simulations, University of Southern California, Los Angeles, CA 90089-0242, USA. [2]Stanford PULSE Institute, SLAC National Accelerator Laboratory, Menlo Park, California 94025, USA. [3]Department of Mechanical Engineering Texas A&M, 400 Bizzell St, College Station, TX 77843, USA. [4]Neutron Scattering Division, Oak Ridge National Laboratory, Oak Ridge, TN 37831, USA. ✉e-mail: chengy@ornl.gov; kolesnikovai@ornl.gov; priyav@usc.edu

thermodynamic behavior, with important implications for their applications in energy, biological, and pharmaceutical systems. Many relevant properties (such as heat capacity, melting, conformation, and polymorph formation) are directly or indirectly driven by molecular vibrations (or phonons in the solid state). The vibration also reflects the local environment of the molecule and the intermolecular interactions (e.g., van der Waals forces and hydrogen bonding). Therefore, it is also crucial for understanding physical and chemical processes such as adsorption and catalysis.

For measuring the full vibrational density of states, inelastic neutron scattering is a powerful tool that can easily access low frequency regions (sub-Thz) and for strongly coherent scatterers the $Q$ dependence of the phonons can easily be obtained[14]. In addition, the calculation of INS is rigorous and straightforward if the dynamics of nuclei can be solved, and explicit treatment of electronic structure is not required. Thus, comparisons between INS, optical spectra, and phonon calculations from atomistic dynamics can provide further insights into electronic and vibrational structure of the molecular system. All these features make INS an appealing technique for studying phonons in molecular solids and liquids.

High-quality INS data is necessary to develop accurate models of the dynamic behavior of molecular solids and liquids, as multiple complicating factors require careful considerations, such as van der Waals interactions, nuclear quantum effects (NQEs), and phonon anharmonicity. Goyal et al. in 1972[15] was able to investigate the inter-molecular spectrum at one temperature in solid phase for ammonia, and Jack Carpenter et al. from 2004[16] measured the density of states at both solid and liquid phases up to 250 meV, but lacks enough energy points for rigorous comparison theoretical models of the fine vibrational structure (especially at inter-molecular energies), and the dispersion ($Q$ dependence) was not measured nor the high energy stretching modes. With modern neutron facilities and advanced simulation techniques, it is now possible to obtain high resolution neutron data along the full range of vibrational energies at multiple temperatures in solid and liquid phase and compare these results to different physical models that can consider van der Waals interactions, NQEs, and phonon anharmonicity on a first principles basis.

In particular understanding the role of van der Waals force is particularly relevant in molecular systems as it is a significant part, if not a dominant part, of the intermolecular interactions[6]. The conventional density functional theory (DFT) cannot describe van der Waals interactions, and empirical corrections are often included, leaving additional uncertainties when modeling such systems. Moreover, most molecular solids contain light elements such as H, for which NQEs could be significant, especially at low temperatures (even though the impact can also be observed at room temperature). Conventional lattice dynamics or molecular dynamics treat nuclei as classical point particles with no spread; thus NQEs are not considered. Last but not least, molecular solids are usually "soft" and tend to exhibit phonon anharmonicity. Such anharmonicity could be coupled with NQEs, making the analysis even more demanding.

Traditionally incorporating all the described effects into one physical model is extremely challenging due to the excessive computational cost. For example, ab initio path integral molecular dynamics (PIMD) simulations[17–20] based on DFT allows one to consider NQEs, phonon anharmonicity, and van der Waals interactions (within the chosen DFT exchange-correlation functional); however, it is extremely costly as it requires multiple replica DFT simulations to be performed. As most of the computational expense for ab initio PIMD simulations comes from having to compute multiple replica DFT simulations, the computational cost can be greatly decreased if the underlying DFT simulations can be replaced by much cheaper computational models.

In this regard, neural-network quantum molecular dynamics (NNQMD) simulations[21] based on machine learning offer a promising tool reduce the computational cost as they revolutionize atomistic modeling of materials by following the trajectories of all atoms with quantum-mechanical accuracy at a drastically reduced computational cost. NNQMD can not only predict accurate interatomic forces but can capture quantum properties such as electronic polarization[22] and electronic excitation[23], thus the 'Q' in NNQMD. A more recent break-through in NNQMD has drastically improved the accuracy of force prediction over those previous models, which was achieved through rotationally equivariant neural networks based on a group theoretical formulation of tensor fields[24–26]. Thus combining PIMD simulations with NNQMD, one can obtain highly accurate first principles based prediction of the INS spectrum.

Here we report measured vibrational density of states and dynamic structure factor for deuterated and protonated ammonia along the solid-to-liquid phase transition with inelastic neutron scattering using SEQUOIA[27] and VISION[28] spectrometers at Oak Ridge National Laboratory, and their comparison to DFT and NNQMD based simulations. Our measured INS spectrum shows strongly anharmonic behavior of the inter-molecular phonon dynamics in solid phase. However, little change in the vibrational spectrum for the intra-molecular modes is observed as a function of temperature in solid phase. In the liquid phase we find hardening of the high energy N-H stretching modes compared to that of the solid phase, which indicates a decrease in the strength of inter-molecular interactions in the liquid phase. We find standard DFT simulations are highly sensitive to the choice of the van der Waals correction to the exchange functional and fail to reproduce the INS spectrum. Through *ab-inito* PIMD and large scale NNQMD-based PIMD simulations we illustrate the discrepancy comes from phonon anharmonicity and its coupling with NQEs. The introduced computational approach to model the INS spectrum is scalable to any material system, offering a robust method to quantify the role of NQEs on material vibrational dynamics.

## Results
### Experimental results
We performed INS experiments to obtain the dynamic structure factor, $S(Q,E)$, as a function of momentum $Q$ and energy $E$ transfer for liquid and solid ammonia using SEQUOIA and VISION spectrometers at Oak Ridge National Laboratories. SEQUOIA is a direct geometry spectrometer that has wide range of energy transfer from sub-meV range up to an eV, with a good energy resolution, thus covering all possible vibrations in solid and liquid ammonia. Neutron scattering angles at SEQUOIA are from 3 deg to 60 deg, thus providing relatively low neutron momentum transfer at high energy transfer, which is very important to measure the intramolecular modes of ammonia, despite the large mean-squared displacement of its hydrogen atoms. VISION is an indirect geometry neutron spectrometer optimized for chemical spectroscopy and molecular systems. It has a wide dynamic range and high resolution, especially in the range of inter-molecular modes for ammonia. Details on the experimental setup are provided in the methods.

Figure 1a, b, shows integrated dynamic structure factor:

$$S(E) = \int dQ S(Q,E) \qquad (1)$$

measured with the SEQUOIA spectrometer for deuterated ammonia ($ND_3$) with incident energies in the range for intermolecular vibrations of $ND_3$. The INS spectra for protonated ammonia ($NH_3$) measured with the VISION spectrometer at base temperature 5 K, and 60 K are illustrated in Fig. 1c. The integrated dynamic structure from SEQUOIA for protonated ammonia is shown in supplemental Fig. 1, which is in good agreement with the measurements taken with VISION. The coherent (incoherent) neutron scattering cross sections for H, D and N atoms are 1.76 b (80.26 b), 5.59 b (2.05 b) and 11.01 b (0.50 b), respectively. For accurate determinations of the phonon density of

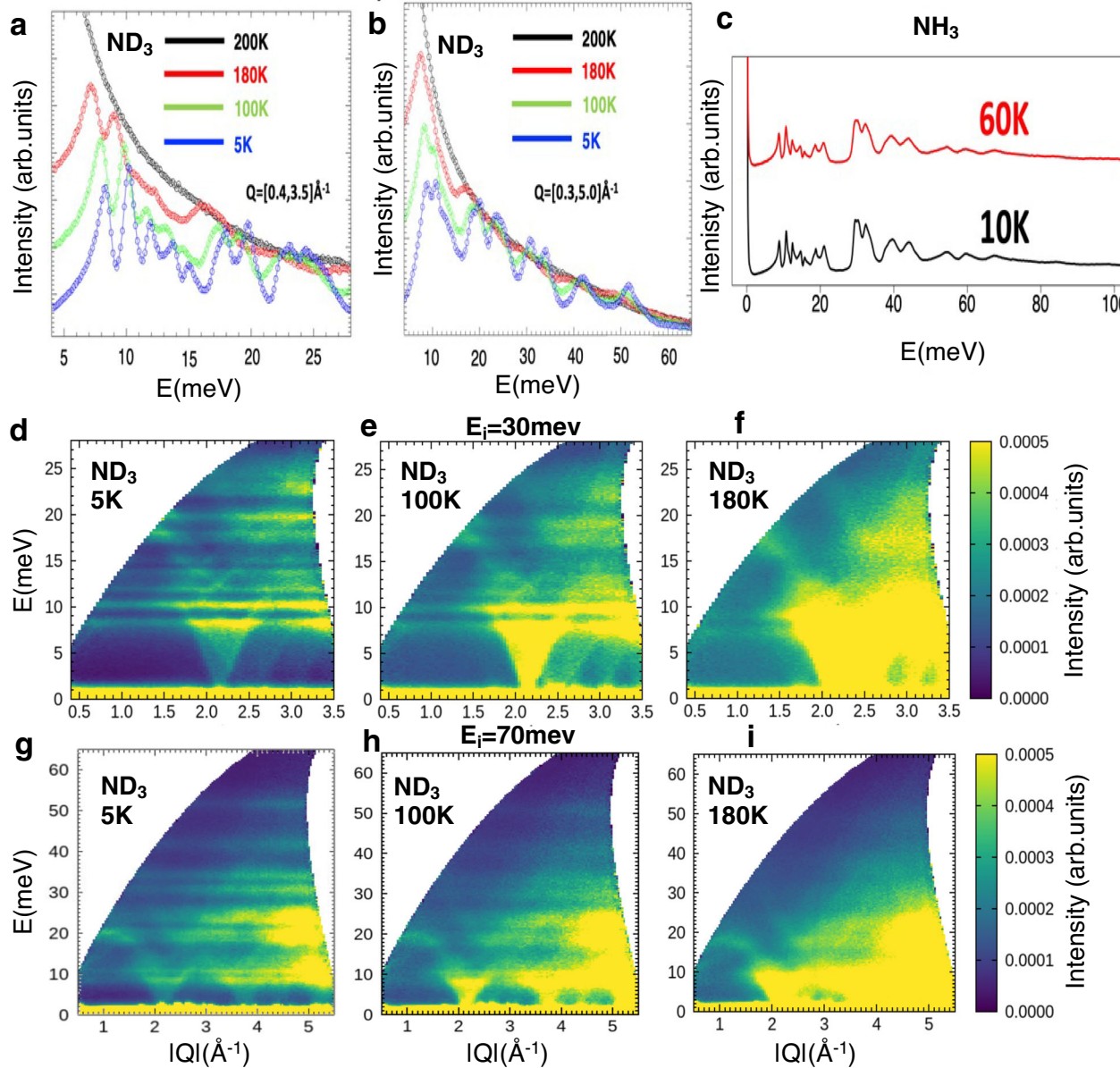

**Fig. 1 | Dynamic structure factor and vibrational spectra for solid and liquid ammonia. a, b** Integrated dynamic structure factor $S(Q,E)$ measured with SEQUOIA spectrometer for incident energies of 30 and 70 meV at various temperatures for $ND_3$ (**c**) Integrated $S(Q,E)$ measured with VISION spectrometer for $NH_3$ (the spectra are vertically shifted for clarity). **d–f** Full $S(Q,E)$ spectra measured for $ND_3$ with incident energy 30 meV with increasing temperature from left to right plotted with same intensity scale. **g–i** Full $S(Q,E)$ measured for $ND_3$ with incident energy 70 meV with increasing temperature from left to right plotted with same intensity scale.

states when the scattering is predominantly coherent, as in the case of $ND_3$, it is very important to average the measured neutron scattering over a large volume of reciprocal space for the resulting data to reflect the true phonon density of states[29]. In the current experiment the ratio of the momentum transfer coverage to the Brillouin zone volume was about 20 for the lowest incident energy of 30 meV (and this value is larger for larger incident energy), therefore the condition of averaging was valid in the whole range of energy transfer studied.

Figure 1a, b, and supplemental Fig. 1 demonstrate that with increasing temperature past 100 K towards the melting point (195 K), peaks in the acoustic and optical regimes display a strong anharmonic softening. At 180 K, majority of features in the spectrum are lost due to anharmonic broadening and Debye-Waller effects. Past the melting transition all prominent features in the vibrational spectrum are washed out.

For $ND_3$, phonon dispersion can be seen in the dynamic structure due to the large coherent cross-section of deuterium and nitrogen atoms. Figure 1d–f, g–i illustrate the dynamic structure factor measured using the SEQUOIA spectrometer with increasing temperature from left to right for incident energies of 30 and 70 meV respectively. At 5 K, acoustic phonon branches can propagate from $Q \sim 2.15$ and $3.75 \, Å^{-1}$, with branch energy ~8.0 meV, and first optical branch is seen at ~10.5 meV. Due to strong phonon anharmonicity, the branches become increasingly smeared and softened with increasing temperature.

We also examined the vibrational energies of the intra-molecular modes as a function of temperature, which are illustrated in Fig. 2a, b for $ND_3$ and 2d, e for $NH_3$. The low energy intramolecular modes do not change as a function of temperature, despite declines in intensity due to Debye Waller effects. Interestingly though for the high-energy N-H stretching modes, we see a merging of the symmetric and anti-

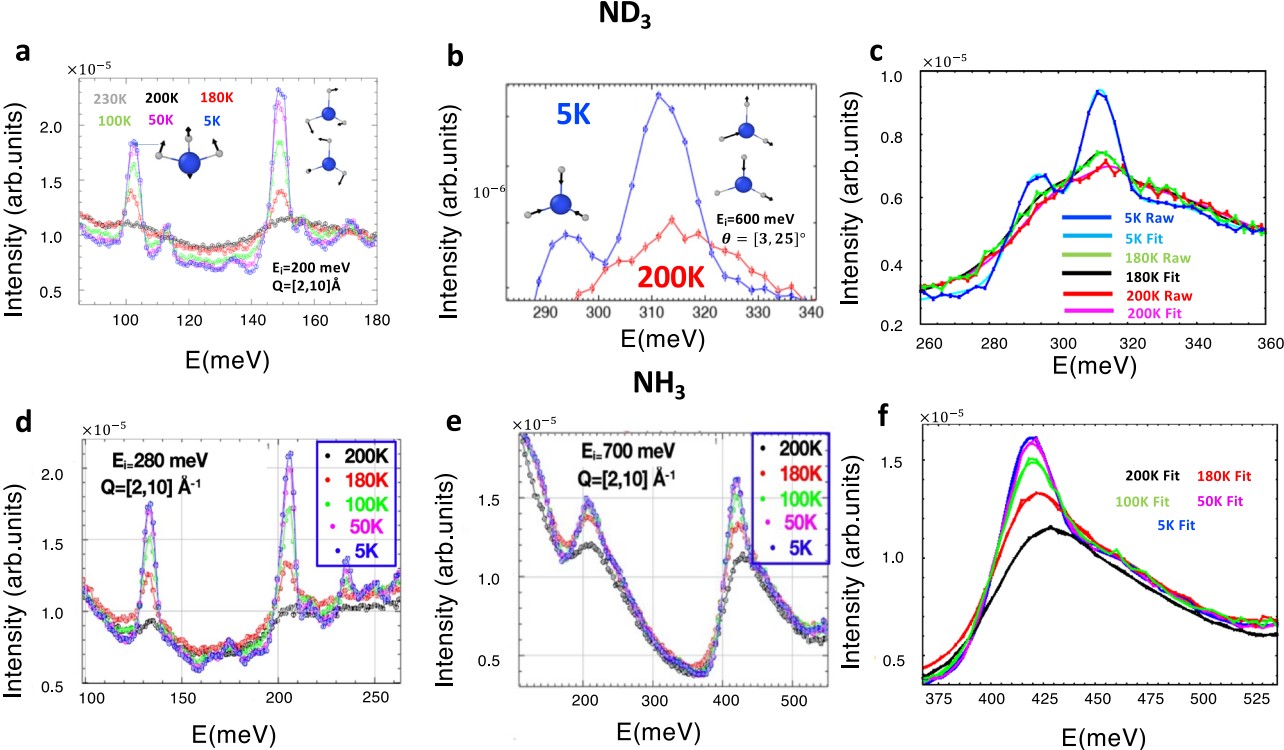

**Fig. 2 | Intramolecular modes for ammonia. a, b** Illustrate molecular modes as a function of temperature measured for ND₃. **c** Illustrate a double Gaussian fit high energy N-D stretching modes. **d, e** illustrate molecular modes as a function of temperature measured for NH₃. **f** Double Gaussian fit for high energy N-H stretching modes, where shift in liquid phase becomes clearer.

symmetric stretching peaks, but this peak split is resolved for ND₃. The merging of the symmetric and anti-symmetric stretching peaks in the protonated sample was determined to be a result of NQEs, which is discussed later. In addition, an increase can be seen in the high energy N-H(D) stretching peak energy in liquid phase compared to the solid phase, which indicates stronger inter-molecular interactions in the solid phase.

To get a quantitative information on the hardening of the N-H(D) stretching peaks as a result of melting, we performed double Gaussian (for ND₃) and single Gaussian (for NH₃, due to non-resolved stretching modes) fits of the neutron data with the addition of linear and Gaussian background terms which are illustrated in Fig. 2c, f for ND₃ and NH₃ respectively. The Gaussian background term is centered at about 450 and 325 meV for NH₃ and ND₃ respectively, and has very large full width at half maximum (FWHM ≈ 73 and 50 meV, for NH₃ and ND₃ respectively), therefore the origin of this peak can be explained by multiphonon neutron scattering involving N-H(D) stretching and low energy intermolecular modes. For ND₃ we have compared the spectra at 5 K and just before and after the melting (at 180 and 200 K), when the phonon/vibrational populations are very similar. For ND₃ we found an increase in $\omega_1 = 293 \rightarrow 297 \rightarrow 301$ meV and $\omega_2 = 312 \rightarrow 313 \rightarrow 316$ meV when transitioning from $T = 5 \rightarrow 180 \rightarrow 200$ K for the symmetric and antisymmetric peaks. For NH₃, little change was seen in the peak energy in solid phase upon further populating the phonons with increasing temperature but the shift in solid to liquid phase was clear with $\omega_{1\&2} = 417 \rightarrow 418 \rightarrow 421$ meV for $T = 5 \rightarrow 180 \rightarrow 200$ K. These results are tabulated in Table 1. It is known that recoil neutron scattering plays a significant role at large $E_i$ in weakly bound molecular crystals[30]. Supplementary Fig. 2 shows that the neutron recoil scattering on ammonia molecule was really observed in the INS spectra measured with $E_i = 700$ and 500 meV, which results in shift of the observed N-H(D) stretching modes (averaged over $Q = 6 - 10$ Å⁻¹) to larger energy by about 8.5 meV

(7.2 meV for ND₃), compared to values of the stretching modes at zero momentum transfer (the values of the stretching modes corrected on the neutron recoil are presented in the supplemental tables 2 and 3). At incident neutron energies $E_i = 280$ meV and below, the neutron recoil scattering on ammonia is insignificant (see Supplementary Fig. 3).

In the liquid phase of ND₃ at 200 K, we also saw coherent excitations in $S(Q,E)$ at low energies resembling acoustic phonons (similar to the solid phase, see Fig. 1d–i) propagating from the first sharp diffraction peak ($Q \sim 2.1$ Å⁻¹), which is illustrated in supplemental Fig. 5. Similar $Q$ dependence of the dynamical structure factor has been observed with inelastic x-ray scattering of water[31].

**Theoretical analysis**

We performed DFT simulations to model the observed INS spectra for solid and liquid ammonia. Figure 3 shows the computed INS intensities for NH₃ based on lattice dynamics and density functional perturbation theory (DFPT) using the Perdew-Burke-Ernzerhof (PBE) implementation[32] of the generalized gradient approximation (GGA) for exchange-correlational functional with various methods of non-local dispersion corrections (see methods for details). These are plotted in comparison with the VISION experiment at 5 K. The OCLIMAX[33,34] software was used to compute the INS intensities from DFT simulations. The simulated spectra strongly depend on the dispersion correction model, and while the overall profile of the spectra seems to be similar, the exact peak positions vary significantly. The variation across different models is unsurprising as the lower energy motions are more sensitive to intermolecular interactions. However, none of the simulated spectra agrees with the experiment well. Comparisons for computed $S(Q,E)$ phonon dispersion curves for ND₃ with SEQUOIA measurement are also illustrated in Fig. 4. For these comparisons; we have also investigated the use of popular hybrid functional B3LYP[35] with inclusion of van der Waals interactions via the DFT-D[36] method, and the meta-GGA SCAN functional[37], with rvv10[38] method for van der

Waals interactions. The scaling parameter in the DFT-D method for the B3LYP-DFT-D simulations was adjusted to give near the experimental lattice constant for the $NH_3$ crystal, similar to other scaling adjustment methods[39]. The hybrid functional performs similarly to the optPBE[40].

**Table 1 | Fitted High Energy N-H(D) stretching modes (in meV) to double (for $ND_3$) and single Gaussian fits (without correction for the neutron recoil shift)**

|        | 5 K      | 180 K    | 205 K    |
|--------|----------|----------|----------|
| $ND_3$ | 293, 312 | 297, 313 | 301, 316 |
| $NH_3$ | 417      | 418      | 421      |

Gaussian fit parameters are described in supplemental figures 9 and 10.

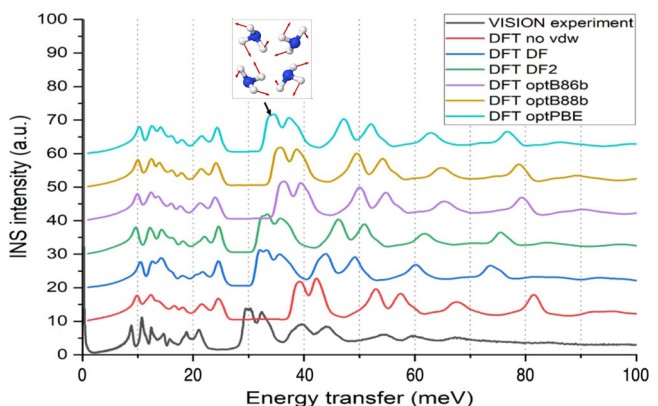

**Fig. 3 | Computed INS spectrum for solid $NH_3$ with various dispersion interactions in comparison to VISION experiment.** Significant discrepancies are observed, especially for the librational band (as marked by the arrow with an illustration of a librational mode in solid ammonia).

van der Waals implementation, which best matched the VISION experiment among the tested models, but the SCAN+rvv10 implementation performs much worse with too-hardened acoustic branches, with much lower relative intensity. The different intensity indicates much different phonon modes predicted by SCAN compared to experiment as the intensity is proportional to the density of phonon states weighted by the atomic squared amplitude of the eigenvectors[33].

One might find this surprising given SCAN's success in describing dynamics of water[22,41,42], and SCAN has also been reported to describe the structure of liquid ammonia compared to elastic neutron experiments[7]. SCAN+rvv10 also under-estimates the 0 K lattice constant with $a^{SCAN+rvv10} = 4.890$ Å. For comparison, we found at 0 K $a^{optPBE} = 5.075$ Å, and we adjusted the scaling parameter in the DFT-D method to give $a^{B3LYP+DFT-D} = 5.033$ Å at 0 K. The measured value at 2 K from neutron diffraction measurement is 5.048 Å[43]. Similar type scaling adjustments have been made for B3LYP with DFT-D and given lattice parameters of $a^{B3LYP+DFT-D} = 4.987$ Å for ammonia crystal[39]. The above values do not include finite temperature and zero-point motion. Within the quasi-harmonic approximation that considers the harmonic zero-point vibrations, the optPBE predicted lattice constant is 5.098 Å at 2 K. A similar small expansion is expected for the other predicted lattice constants.

For SCAN it was also found that "harder" (i.e., lower cutoff radius) GGA pseudo-potentials than typically used were needed. For softer pseudo-potentials, much worse performance and an unphysical hypersensitivity of the phonon modes to the density was found. This is most likely due to issues of transferability of GGA pseudo-potentials to meta-GGA functionals, which is still an open issue in the DFT community[44,45]. All the above issues were seen in both SCAN and SCAN+rvv10; however, SCAN without van der Waals correction did report a better agreement with the lattice constant ($a^{SCAN} = 4.956$ Å).

While optPBE and the hybrid functional perform better in comparison to SEQUOIA measurement than SCAN for $ND_3$, they still slightly overestimate optical modes, especially in the 25–55 meV range.

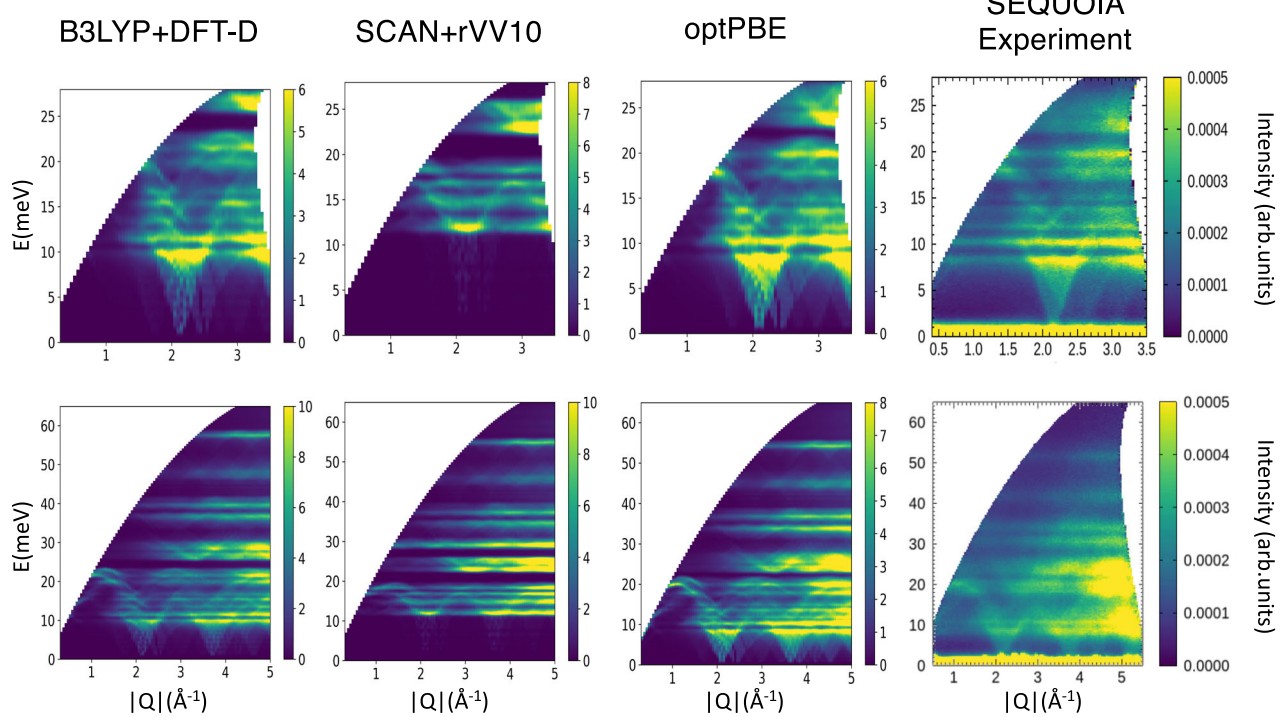

**Fig. 4 | Computed $S(Q,E)$ with various exchange-correlation functionals and dispersion interactions for solid $ND_3$ in comparison to SEQUOIA experiment ($T = 5$ K).** The agreement of DFT simulated and INS spectra for $ND_3$ is better than for $NH_3$ indicating the role of nuclear quantum effects.

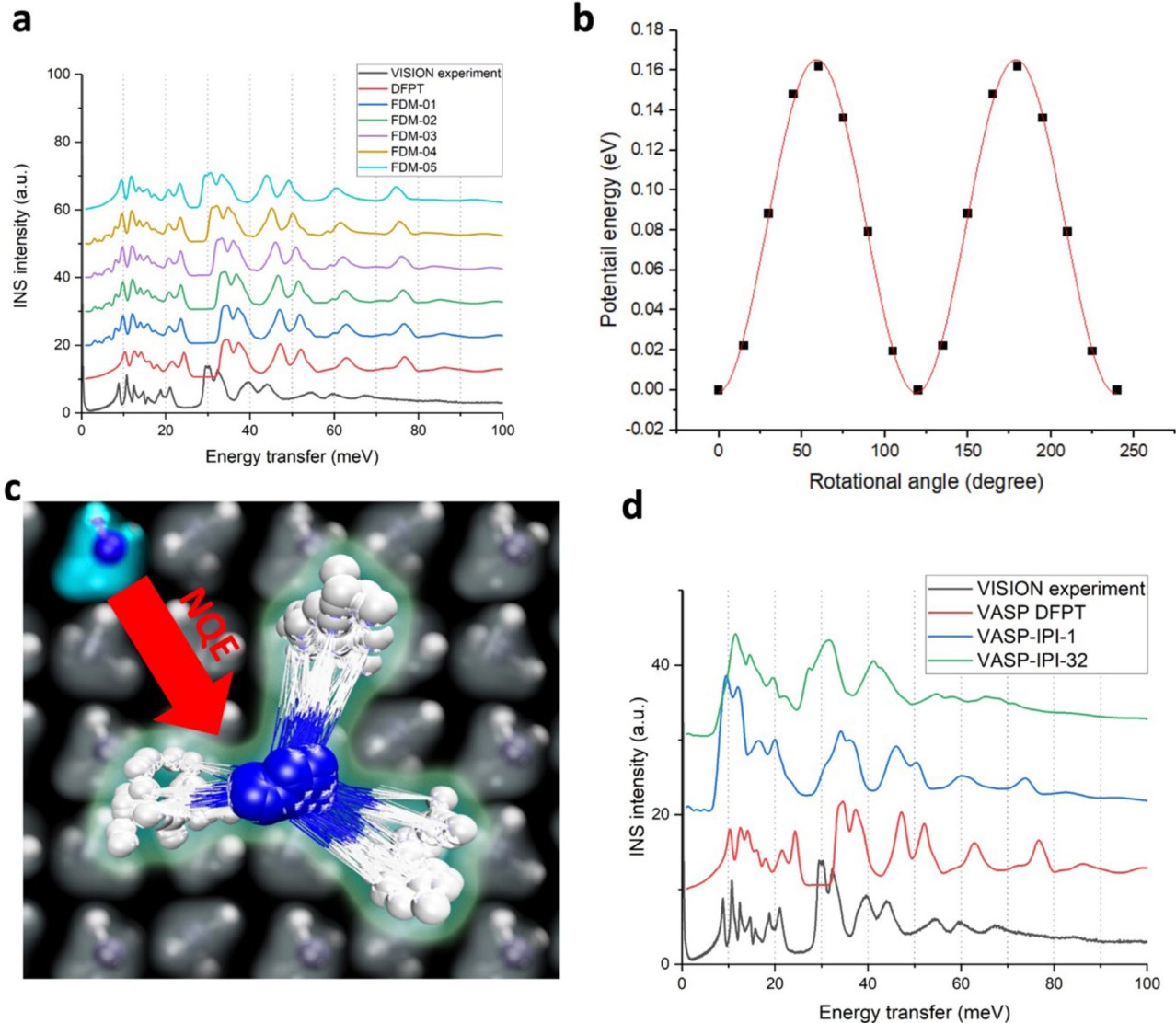

**Fig. 5 | Evaluation of nuclear quantum effects with DFT. a** INS spectra of solid ammonia simulated with various step sizes using the finite displacement methods (e.g., FDM-01 means a step size of 0.1 Å). The drifting librational band is an indication of its anharmonicity. **b** Potential energy profile of $NH_3$ rotation in solid ammonia, solved by nudged elastic band method. The energy barrier is determined to be 167 meV and the associated $NH_3$ quantum rotor has an excitation energy ($n = 0 \to 1$) of 32 meV. **c** Illustration of PIMD simulation. While typical ab initio simulation treats atoms classically while electrons quantum-mechanically, PIMD simulation uses multiple replicas of each atom to mimic nuclear quantum effect. **d** INS spectra of solid ammonia from DFPT, ab initio PIMD (1-bead and 32-bead TRPMD) simulations, and VISION experiment. The PIMD simulation with 32 beads reproduces the experimentally observed peak positions very well, whereas the other two exhibit major discrepancies.

We also found that optPBE computed radial distribution functions in liquid state illustrate good agreement with previous neutron experiments extracted from refs. 4,46 (see Supplementary Fig. 6), especially the N-N distribution function which arises from inter-molecular interactions. Thus, the optPBE functional adequately describes the structure of the liquid and solid state, but fails to capture the inter-molecular vibrational spectrum, indicating something still needs to be added that goes beyond the choice of exchange functional. By visualizing the vibrational modes for $NH_3$, it is found that the peaks at ~40 meV in simulation (Fig. 3) are due to $NH_3$ librational modes, i.e., rotation of $NH_3$ molecules around the "umbrella" axis. This presumably corresponds to the peaks at ~30 meV in the VISION experiment. Such a significant difference (~30%), regardless the van der Waals model used, is highly unusual for such a seemingly simple crystal. It points two possibilities: 1) any available model cannot describe the true van der Waals interaction; or 2) these modes are highly anharmonic thus, the harmonic approximation as used by DFPT failed. Since the discrepancy

is seen with the data obtained at 5 K, the anharmonicity is caused by extension of the atomic phase space due to zero-point motion of the atoms rather than their thermal motion. This hypothesis would also be consistent with the fact that while there is some discrepancy with the predicted $S(Q,E)$ spectrum for $ND_3$, it is smaller than that observed with $NH_3$ measurements at VISION due to the heavier mass of deuterium.

To evaluate the contribution of anharmonicity, finite displacement method (FDM) was used to calculate the phonons. Unlike DFPT, which measures the second order derivative at the bottom of the potential energy basin, FDM can probe the vicinity by adjusting the stride size. Interestingly, Fig. 5a shows that with larger displacement, the frequency of the librational band has a significant and systematic redshift, closing the gap between the experiment and simulation. This is a clear indication that the librational modes are anharmonic, and the corresponding potential energy profile is non-parabolic, but the displacement method is not meant to be quantitatively accurate with 0.5 Å being much larger than the expected mean-square displacement

of the atoms at 5 K and accordingly the other parts of the spectra do not show a consistent improvement.

The anharmonicity of a particular mode (in this case, the $NH_3$ libration) can also be evaluated by mapping out the potential energy profile corresponding to the mode, which can be obtained by nudged elastic band calculations. The result in Fig. 5b shows that the three-fold potential energy profile has a barrier of about 167 meV. As a quantum rotor in this potential well, the excitation energies can be predicted using a quantum rotor model implemented in DAVE[47], which are illustrated in supplemental Table 1. The first two energies ($-1.3 \times 10^{-6}$ meV) are the tunneling splitting of the rotational mode which is beyond our instrument resolution. The next three energies (32 meV) correspond to the excitation of the rotational vibration of $NH_3$ (corresponding to $n = 0 \rightarrow 1$ excitation in a quantum oscillator model). This energy represents what is directly measured in our INS experiment and is good agreement with the VISION measurement for this mode.

The above analysis highlights the problem but has yet to offer a general solution, as the quantum rotor model cannot be used to simulate the entire INS spectra or easily generalized to study other modes/systems. While conventional molecular dynamics can accurately describe anharmonic effects due to finite temperature, it cannot capture anharmonic shifts due to the zero-point motion of the atoms which is what captured by the quantum rotor model $n = 0 \rightarrow 1$ excitation. A promising solution is path integral molecular dynamics (PIMD), in which the quantum partition function is mapped to a classical analogue by using replicas (beads) connected by springs (ring polymers) to represent each atom[17], which effectively extends the phase space of the atomic system due to their zero point motion. This method is suited for problems when the particle zero-point energy is not negligible with respect to the average thermal energy, which is expected for hydrogen systems at low temperature such as ammonia. The background of Fig. 5c shows a typical first principles-based simulation, where the atoms are treated classically, and the electron charge density is treated quantum-mechanically to compute atomic forces, illustrated as gray iso-surfaces. In the foreground, we have highlighted one $NH_3$ molecule from a PIMD simulation of the same atomic configuration, where each atom has 32 replicas that are harmonically coupled together. As we increase the number of frames visualized, the beads effectively represent the phase space visited by the nitrogen and hydrogen atoms due to both there thermal and zero-point motion. Supplementary Fig. 4 shows bead positions for one ammonia molecule for 1000 frames in an ab-initio PIMD simulation illustrating the large phase space visited by the light hydrogen atoms.

The computation of the replica simulations is embarrassingly parallel, with only fixed nearest replica communication, and the major cost is computing the energy and forces for the atoms within each replica simulation, which is done from first principles. Different flavors of path integral implementations exist, but not all are suitable for spectroscopic modeling through time correlation functions. For example, two implementations of PIMD, ring polymer molecular dynamics[19] (RPMD) and centroid molecular dynamics[18] (CMD), have been tested to simulate infrared spectra. It is found that high frequency modes tend to suffer from artifacts: the resonance problem in RPMD shows spurious peak splitting and the curvature problem in CMD shows red-shift and broadening of the stretching peak. A recently proposed method of adding a thermostat to the conventional RPMD[20] (TRPMD) provides a potential solution to both issues. In Fig. 5d, four spectra are compared, the experimentally measured INS spectrum, the simulated INS spectrum using DFPT, and two spectra simulated from TRPMD, with 1 bead (equivalent to conventional molecular dynamics without NQE) and 32 beads, respectively. The DFPT spectrum and the 1-bead TRPMD spectrum are similar; however, there is a significant loss in resolution in the simulated curve due to the small sample size and limited run-time available with *ab intio* MD. The simulated spectrum from 32-bead TRPMD looks very different and is much closer to the

experimentally measured spectrum. Specifically, the position of the librational band and the peaks at higher energies, are all in much better agreement with the experiment. This lack of resolution leaves much to be desired when trying to conclusively determine the majority of spectrum discrepancies arising from NQEs.

As most of the computational expense for *ab intio* PIMD simulations comes from having to compute multiple replica DFT simulations, the computational cost can be significantly decreased if the underlying DFT simulations can be replaced by much cheaper computational models. To circumvent this issue, we performed NNQMD based TRPMD simulations utilizing the recently developed group-theoretically equivariant neural-network forcefield model Allegro[26], whose architecture is illustrated in Fig. 6a. Allegro falls under the class of graph-based models where each atom $i$ represents a node in the graph and edges are the interatomic distances $r_{ij}$, while retaining data locality of features to achieve high computational speed[24]. For each node $i$ the model is designed to predict the edge/pair energies $E_{ij}$, and the total energy is the sum of all pair energies for each node. Forces are then computed through gradients of the total energy.

Based on the work of the state-of-the-art neural equivariant interatomic potential (NEQUIP)[25], Allegro utilizes equivariant features in the form linear embeddings of spherical harmonic projections of pair-wise atomic distances ($r_{ij}$). The model is "equivariant" in that only tensor operations on spherical harmonic projectors that respect the symmetry of the Euclidean group $E(3)$ are performed. Allegro also allows for scalable and faster computations compared to previous works as the model is local, with spherical harmonic embeddings only performed on neighbors within a cutoff distance. For further details, the reader is directed to the methods and to Ref. 26. Training for the model was done using configurations from solid and liquid DFT-MD simulations and configurations from the PIMD simulations. Details on the model training are provided in the methods.

With this framework, we performed large-scale TRPMD simulations in the crystalline phase in a ~ 30 Å box consisting of 864 $NH_3$ molecules (3456 atoms) at 60 K. Figure 6b shows the measured spectrum on VISION spectrometer at 60 K and those computed with TRPMD utilizing the allegro model which shows excellent agreement. The 0 K DFT calculation which over-estimates the high energy optical modes, is also shown for perspective. In addition, in Fig. 6c, we computed the vibrational density of states for the high energy N-H stretching using classical MD and TRPMD from Fourier transform of velocity auto-correlation function, which demonstrates the merging of the symmetric and anti-symmetric modes seen in the SEQUOIA experiment is due to anharmonic broadening from zero-point motion. Further, we can show the extended tail seen in the SEQUOIA experiment is due to multi-phonon neutron scattering (stretching plus low energy modes) through computation of INS spectrum within the incoherent approximation with OCLIMAX software which is illustrated in Fig. 6d. In addition, we found computation for $ND_3$ vibrational spectrum did not show this peak merging and is presented in Fig. 6e. This is consistent with SEQUOIA measurement and further validates the role of zero-point motion in the anharmonic broadening of these modes.

We also examined the dynamics in liquid phase through PIMD simulations of 864 $NH_3$ molecules in a cubic simulation box of size 32 Å at 205 K. We computed the vibrational density of states via Fourier transform of the velocity autocorrelation function for liquid $NH_3$ which is illustrated in Fig. 6f. As seen with SEQUOIA experiment the low-frequency intermolecular modes are all washed out and the computed values for the molecular peaks are in good agreement with the measured values. Figure 6g shows the computed INS spectrum including multi-phonon scattering for solid and liquid $NH_3$, where we also see a hardening in high energy N-H stretching modes compared to those computed in the crystalline phase, which indicates weaker intermolecular interactions in the liquid. Care must be taken when

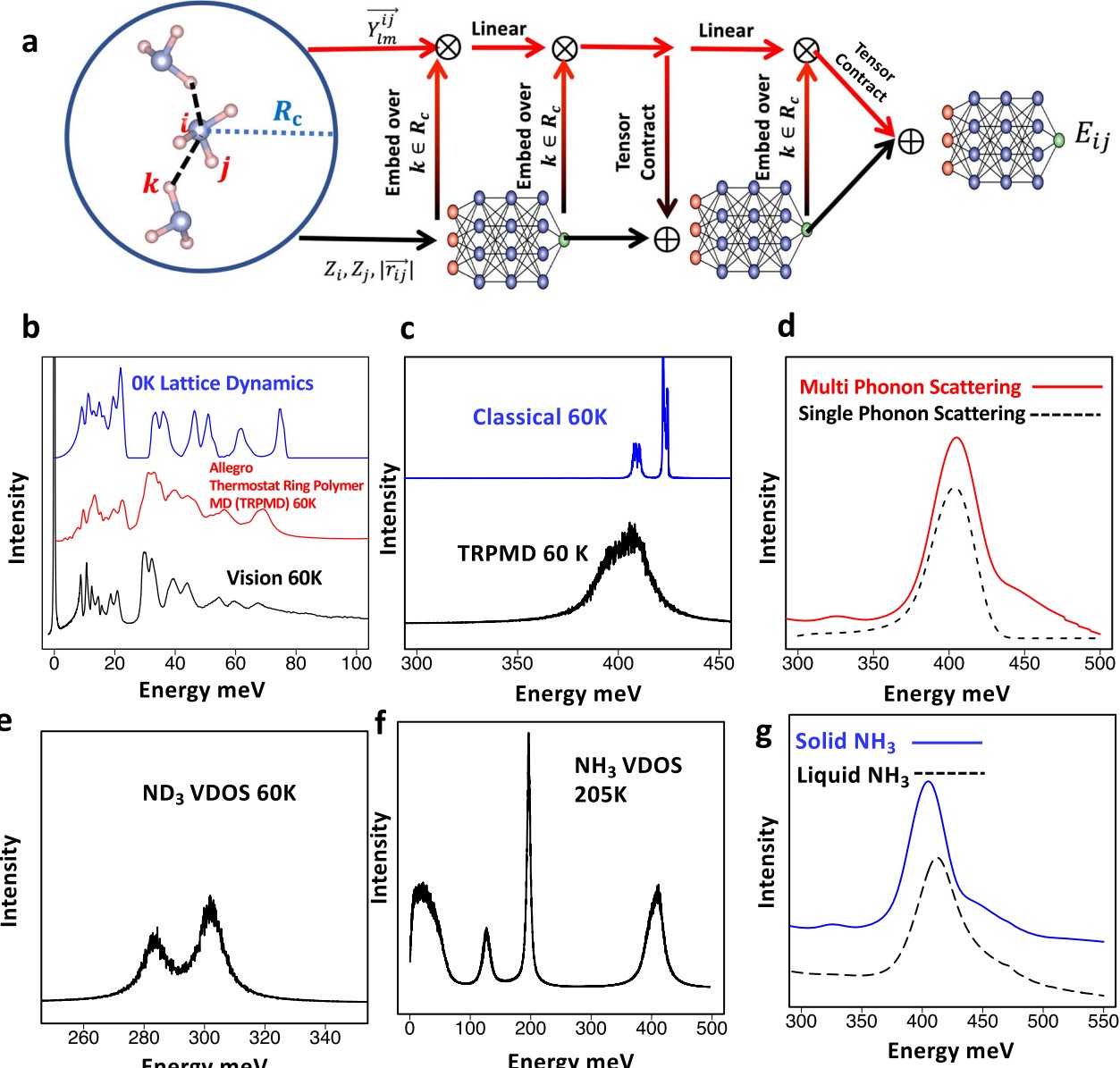

**Fig. 6 | TRPMD simulations of ammonia using NNQMD. a** Allegro model for NNQMD. **b** Comparison of computed vibrational spectrum using Allegro TRPMD and VISION spectrum which shows excellent agreement. DFT results at 0 K (blue color) are shown for comparison. **c** Comparison of classical and TRPMD simulation of high-energy N-H stretching modes. Merging of Symmetric and Anti-symmetric splitting agrees with SEQUOIA experiment. **d** INS computation of spectrum near

N-H stretching peak from summing only single phonon scattering and with inclusion of multi-phonon scattering. **e** TRPMD $ND_3$ VDOS for high-energy N-D stretch where peak splitting remains. **f** Computation of VDOS for $NH_3$ liquid state, in which low energy peaks are all washed out. **g** Computed hardening of N-H stretching modes of $NH_3$ in liquid state consistent with SEQUOIA experiment.

incorporating the multi-phonon shifts within the harmonic approximation in the so called "Gaussian approximation"[48], especially in the liquid phase. In OCLIMAX we used a direct convolution of the low- and high-energy modes (up to 10 orders, but still in harmonic approximation) and as a result the overall agreement is still reasonable with experiment. The good agreement in liquid is largely due to the phonon structure in liquid phase at low energy both computed and measured is highly unstructured resulting in a unstructured multi-phonon background. As this frequency shift signifies stronger inter-molecular interaction in the solid phase, which one typically attributes to hydrogen bonding, we examined the charge density overlaps in the two different phases. From DFT-MD simulations, the threshold for charge density overlap in crystalline phase was found to be on

contours of 0.013 e/$Å^3$, whereas in the liquid phase molecules could be found sharing charge density iso-surfaces at almost twice the value seen in the solid phase (see Supplementary Fig. 7). Stronger charge density overlap in liquid would normally indicate stronger intermolecular interactions in the solid phase. However, lifetime of the strong-intermolecular interactions with large density overlap in the liquid phase (i.e., hydrogen-bond lifetime) is extremely short in liquid ammonia, with reported hydrogen-bond lifetimes in liquid ammonia have been on the order of ~100 fs[4,7]. Thus, in the liquid phase there can be instantaneous periods of strong inter-molecular interaction between a few molecules, but this interaction is transient. In contrast in the solid phase, there is temporally constant weak interaction of the molecules. This, on average constrains them more resulting in a softer

stretching vibration, which is reflected in the measured and computed stretch peaks in the spectra.

## Discussion

We have performed inelastic neutron scattering (INS) measurements on solid and liquid ammonia and compared the measurements to density functional theory (DFT) simulations. We find nuclear quantum effect (NQE) induced anharmonicity that fundamentally changes the predicted spectrum with conventional DFT simulations, which we illustrate through neural network-based path integral molecular dynamics (PIMD) simulations using the thermostat ring polymer molecular dynamics (TRPMD) implementation of PIMD. PIMD simulations can reproduce the hardening of N-H stretching modes in the liquid phase. The hardening was determined to be due to different spatial and temporal characters of the hydrogen bonds. In solid phase, the constant and percolated hydrogen bonding network makes the N-H stretching modes softer than in the liquid phase, where brief periods of strong inter-molecular interaction are followed by periods of low/non-interaction. The reported solid phase INS measurements on density of states is in good agreement with that by Goyal et al. in 1972, but with much enhanced resolution[15].

In comparisons with optical spectroscopy methods, we find them in general good agreement with the INS data reported here[2,49,50]. For the inter-molecular modes in solid phase, we find good qualitative agreement with peak positions and major features of the density of states, in particular the large gap between the translational and rotational bands at 32 meV, which we demonstrate is heavily influenced by the zero-point motion of atoms in crystal. For the intra-molecular spectrum in both solid and liquid phase we find the neutron spectrum for the high energy stretching modes after correction for the neutron recoil are also in agreement with optical measurements. A table comparing these results is provided in supplementary tables 2, 3.

NQE-induced anharmonicity is expected to be quite common in molecular solids/liquids, especially in systems with "flexible" groups such as -OH, $-CH_3$, $-NH_2$. This is especially relevant for neutron spectroscopy, which typically measures materials with large concentrations of hydrogen at low to intermediate frequency range and at low temperatures. Significant discrepancies between simulated peak positions (harmonic approximation) and measured ones have been observed in many related systems, including ice, amino acids, drugs etc[51–55]. Interpretation of the discrepancies, when comparing theory and experiment, can be complicated and potentially misleading. For example, one can conceivably adjust the parameters in the dispersion correction or exchange-functional so that the simulated INS spectrum matches the experimental one. Still, if the simulation was done without considering NQE and/or anharmonicity, the agreement can be fortuitous and does not reflect the true intermolecular interactions. With the recent explosion of machine-learning-based interatomic potentials, evaluation of NQEs in large spatiotemporal simulations at near DFT accuracy is possible, as demonstrated in a recent study of water/ice where NQEs are found to have a crucial contribution to the stability of ice Ih[56]. The framework presented here is anticipated to serve as a guide for proper analysis of INS experiments in the future. The major goal of spectral analysis is not to make a model that can replicate the spectrum but to have a model that can simulate the spectrum based on the correct physics and thus also have extrapolation power by capturing the fundamental interactions and nuclear quantum dynamics. This is extremely relevant for systems like ammonia, where new technologies and applications such hydrogen storage will need large-scale simulation frameworks such as NNQMD simulations performed here.

## Methods

For the SEQUOIA experiment the protonated ammonia was condensed at 77 K in an aluminum cylindrical annular container of 18 mm outer

diameter, 0.2 mm thick and 50 mm high (see details in supplementary information figure 8). The thickness for deuterated ammonia was 0.7 mm. Each sample was cooled to a base temperature of 5 K and data were collected for several different incident energies. The samples were then heated to several subsequent temperatures for which the same incident neutron energies were used. The INS spectra for empty containers at the same incident neutron energies and temperatures were also measured and subtracted from the sample data.

For VISION experiment, ammonia was condensed and solidified in an aluminum sample holder at 77 K in liquid nitrogen. It was then loaded to the instrument and cooled to the base temperature of 5 K. The INS spectra were then collected at a series of temperatures including 5 K and 60 K.

Density Functional Theory (DFT) calculations were performed using the Vienna Ab initio Simulation Package[57,58] (VASP). Calculations used the Projector Augmented Wave (PAW) method[59,60] to describe the effects of core electrons. Perdew-Burke-Ernzerhof[32] (PBE) implementation of the generalized gradient approximation (GGA), metaGGA SCAN[37], and hybrid B3LYP[35] exchange-correlation functionals were used. The energy cutoff was 800 eV for the plane-wave basis of the valence electrons. The lattice parameters and atomic coordinates from literature[43] were used as the initial structure. For the SCAN functional, hard versions of the GGA PAW potentials provided in the VASP package were required, and a 1000 eV cutoff was used in the plane-wave basis. The unit-cell ground state structure was calculated on a $11 \times 11 \times 11$ Γ-centered mesh for the unit cell (16 atoms). The total energy tolerance for electronic energy minimization was $10^{-8}$ eV, and for structure optimization, it was $10^{-7}$ eV. The maximum interatomic force after relaxation was below 0.001 eV/Å. Dispersion corrections with various models[40,61–63] (e.g., optPBE-vdW[40] functional) were applied. A $3 \times 3 \times 3$ supercell (432 atoms) on Γ-point only was used for the calculation of phonons, using two different methods, DFPT and FDM. The vibrational eigen-frequencies and modes were then calculated by solving the force constants and dynamical matrix using Phonopy[64]. The OCLIMAX[33,34] software was used to convert the DFT-calculated phonon results to the simulated INS spectra.

DFT-based PIMD simulations (with TRPMD) were driven by i-Pi[65], with the interatomic forces calculated by VASP (using optPBE-vdW[40] functional) on a $2 \times 2 \times 2$ supercell (128 atoms). The simulations were performed with 1 bead (classical limit) and 32 beads, with a timestep of 0.25 fs and 40,000 total steps (10 ps trajectory). A 64-bead model (with shorter run time) was also tested for convergence. The temperature for the simulation was 60 K, and the reason for not running at base temperature (10 K) is because the number of beads needed for convergence is much higher at 10 K, making the simulation unfeasible.

Training for the Allegro neural network force-field was done using frames from DFT-MD simulations for the liquid (205 K) and solid phase (60 K) and the above PIMD simulations. These were performed on supercells containing 108 $NH_3$ molecules (432 atoms) using the optPBE-vdW[40] functional with box lengths of 16.2 Å and 15.39 Å respectively. A distance cutoff of 5.0 Å was used to train the model. Allegro implements equivariant features as spherical harmonic projections of atomic pair wise distances ($r_{ij}$). Allegro utilizes two latent spaces: one for equivariant tensor features ($Y_{lm}$), and one for scalar invariant features $r_{ij}, Z_i, Z_j$, with $Z_i$ being the atomic number of species $i$. The equivariant latent space is updated through equivariant tensor products up to a maximum order $l$ with a weighted embedding of spherical harmonics projections of atoms within the central atom's cutoff radius. The weights are determined by a multi-layer perceptron acting on the scalar latent space features. The equivariant features of the same irreducible representation are linearly mixed. Then the scalar latent space is updated through tensor contraction of scalar and new linearly mixed equivariant features which is then passed through a multi-layer perceptron. This process is repeated for $N$ layers and then passed through a final multi-layer perceptron for pair energy

prediction. for *N* layers and then passed through a final multi-layer perceptron for the pair energy prediction. For more details on the model, we refer the reader to reference[26] and https://github.com/mir-group/allegro.

Large-scale NNQMD-based MD and PIMD simulations were performed in supercells containing 864 molecules with box sizes of 30.24 Å and 32.44 Å for solid and liquid phase at 60 K and 205 K, respectively. A time-step of 0.25 fs was used, and 20,000 MD steps (5 ps) of thermalization was performed before production runs of 40 ps (160,000 MD steps). Initialization of liquid structures was taken from replicating already melted simulations from DFT-MD. PIMD simulations were driven by iPi using the TRPMD implementation PIMD. Both the LAMMPS[66,67] and RXMD[68] software were used as the molecular force engines. The Liquid Lib Library[69] was used to compute the velocity auto-correlation functions.

## Data availability
The neutron data, neural network training data, neural network model, DFT structure have been deposited into a zenodo database[70]. (https://zenodo.org/records/10840577.)

## Code availability
All codes used for DFT, MD, PIMD, and machine learning training data generation and prediction are open source or commercially available.

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

## Acknowledgements

This manuscript has been authored by UT-Battelle, LLC under Contract No. DE-AC05-00OR22725 with the U.S. Department of Energy. The United States Government retains and the publisher, by accepting the article for publication, acknowledges that the United States Government retains a non-exclusive, paid-up, irrevocable, world-wide license to publish or reproduce the published form of this manuscript, or allow others to do so, for United States Government purposes. The Department of Energy will provide public access to these results of federally sponsored research in accordance with the DOE Public Access Plan (http://energy.gov/downloads/doe-public-access-plan).

This work was supported in part by the "Scattering and Instrumentation Sciences Program" of the Department of Energy, Office of Science, Basic Energy Sciences, under Award Number DE-SC0000267409. A portion of this research used resources at the Spallation Neutron Source, a DOE Office of Science User Facility operated by the Oak Ridge National Laboratory (ORNL). The computing resources for ab initio PIMD simulations were made available through

the VirtuES project, funded by Laboratory Directed Research and Development (LDRD) program and Compute and Data Environment for Science (CADES) at ORNL. Y.Q.C. is partly supported by the Artificial Intelligence Initiative as part of the LDRD program of ORNL, managed by UT-Battelle, LLC, for the US Department of Energy under contract DE-AC05-00OR22725. T.M.L and Y.Q.C. thank Simon Batzner and Albert Musaelian for their assistance in using NEQUIP and Allegro models. We thank Victor Fanelli and Mark Loguillo for help with the design of the sample container.

## Author contributions

T.M.L., A.K., A.N., A.I.K., and P.D.V. designed and performed the SEQUOIA experiment. Y.Q.C, A.J.C, and L.L.D. designed and performed the VISION experiment. W.R.H. designed the sample container for SEQUOIA experiment. T.M.L., A.K., and Y.Q.C. performed the static DFT simulations and phonon calculations. Y.Q.C. performed the DFT based PIMD simulations. T.M.L. and K.N. trained the Allegro NNQMD model. T.M.L. performed the NNQMD based classical M.D. and PIMD simulations. T.M.L. and Y.Q.C. wrote the first draft of the manuscript. All participated in the data analysis and final editing.

## Competing interests

The authors declare no competing interests.
