## [Peer Review File · Nature Communications]

Neutron scattering and neural-network quantum molecular dynamics investigation of the vibrations of ammonia along the solid-to-liquid transitionREVIEWER COMMENTS

Reviewer #1 (Remarks to the Author):

The manuscript NCOMMS-23-38777, entitled "Neutron Scattering, Neutron Quantum Effects and Neural Networks: The Delicate Case of the Ammonia Vibrational spectroscopy" by T. M. Linker et al., submitted for publication on "Nature Communications", deals with a detailed and accurate study of the inelastic neutron scattering (INS) spectra of solid and liquid ammonia (both protonated NH₃ and perdeuterated ND₃) in the broad temperature range 5-200 K, crossing the sample melting at 195.4 K. The performed investigation is carried out following two distinct but complementary lines: experimental measurements of the INS response making use of two high-level neutron spectrometers (SEQUOIA and VISION at the Spallation Neutron Source); 'ab-initio' DFT simulations of solid and liquid ammonia including anharmonic and quantum effects in the sample microscopic dynamics via an extended use of the so-called Ring Polymer Molecular Dynamics (RPMD) algorithm.

This paper is clearly written, scientifically sound, and provides new and noteworthy results about the ammonia vibrational dynamics which are relevant both from the fundamental and the applied point of view, since NH₃ is a crucial compound in various scientific and technological areas. In addition, it might also represent a sort of benchmark for the analysis and the interpretation of high-resolution INS spectra in all the situations where standard (pseudo-harmonic) lattice dynamics cannot provide a good reconstruction of the measured data. Although this approach is not totally new (see e.g., L. del Rosso et al., 2021, DOI: <https://doi.org/10.1021/acs.jpcc.1c07647>), we have to frankly admit that the present study is able to reach an unprecedented level of accuracy and completeness so to include the full vibrational spectra of the investigated molecules, ranging from low-energy lattice phonons to high-energy stretching modes. This is for sure an original and outstanding result, especially because it is coupled with an accurate and complete experimental data reduction procedure.

My main criticism about this study concerns the absence of high-resolution optical (Raman/IR) spectroscopic data on solid and liquid ammonia (e.g., Binbrek and Anderson, 1972, DOI: [https://doi.org/10.1016/0009-2614\(72\)80205-7](https://doi.org/10.1016/0009-2614(72)80205-7)) which, on the contrary, are very useful, especially for understanding the internal vibrations where the dispersion in Q does not play a fundamental role. The authors should include this information in their discussion, typically comparing INS and optical frequencies. This is particularly important since, exploiting these data, Zeng and Anderson in 1990 set up a complete (although approximate) force-field model of the vibrational dynamics in solid NH₃ and ND₃ (DOI: <https://doi.org/10.1002/pssb.2221620107>).

In addition, some minor points should be considered in the amended version on the present manuscript:

- 1) The experimental INS data on solid ammonia at 106 K reported by Goyal et al. (1972) should be explicitly cited (DOI: <https://doi.org/10.1002/pssb.2220500232>).
- 2) Page 5, lines 1-7: Dealing with quantum effects and anharmonic effects in the ammonia microscopic dynamics there are some statements which sound slightly vague and hand-waving. I think that this theoretical explanation should be surely short but also rigorous: pseudo-harmonic lattice dynamics is exact from a quantum point of view, but it needs that the interparticle potential can be well approximated by a harmonic term. Since it is not often the case, one speaks of "anharmonic effects", which are relevant when the particle mean square displacement is large (i.e., at high temperature or for low particle masses). On the contrary, Newtonian molecular dynamics deals with anharmonic potentials very easily, but it is unable to include "quantum effects" since it is purely classical. Quantum effects are relevant when the particle zero-point energy is not negligible with respect to the average thermal energy (i.e., at low temperature or for low particle masses) This is exactly the reason why one switches to RPMD which can deal with both effects, provided that quantum effects are not too strong.
- 3) Page 7, Eq. (1) and following lines: The use of the Q integration to get rid of the coherent components in the ND₃ INS spectra in order to establish a comparison with the INS spectra from NH₃ is a very common practice and is based on the so-called "Bredov approximation" (see, e.g.,

the study by N. Breuer, 1974, DOI: <https://doi.org/10.1007/BF01677937>) which, however, is rigorously proven only in the case of simple Bravais systems. I think that the authors should briefly mention this issue.

4) Pages 8 and 9: In connection with reconstruction of the INS spectra (starting from the velocity autocorrelation functions calculated via RPMD), where both fundamental and multiphonon contributions are present, the authors should clearly state that their procedure (embedded, for example, in the formulas used by the OCLIMAX code) is exact only in an essentially harmonic context. If applied to anharmonic systems (including, e.g., liquids) this approach is well-known under the name of "Gaussian approximation" (see Rahman et al., 1962 DOI: <https://doi.org/10.1103/PhysRev.126.986>). Especially in the case of liquid ammonia at 200 K, additional care is needed to implement such a multiphonon calculation, since, rigorously speaking, the standard Debye-Waller coefficient does not exist any longer and the self diffusion coefficient causes a noticeable peak broadening, especially at large Q values.

For all reasons mentioned above I think that this work should be published on "Nature Communications", but only after the minor revisions suggested in this review.

Reviewer #2 (Remarks to the Author):

Review on the paper entitled « Neutron Scattering, Nuclear Quantum Effects and Neural Networks: The Delicate Case of the Ammonia Vibrational Spectrum »
by T.M. Linker et al.

The paper presents a study of vibrational modes in ammonia studied by inelastic neutron scattering and numerical modelling. Ammonia is a well studied system, however because of its importance in industry and potentially in energy, it deserves further and in-depth studies. The vibrational properties are key properties since it is related to many thermodynamics properties.

NH₃ and ND₃ were characterised by INS, providing beautiful spectra. Two spectrometers were used, SEQUOIA and VISION in the Oak Ridge neutron center (SNS), although it is not clear what the information obtained on VISION brings after the measurements performed on SEQUOIA. Strong anharmonicity can be straightly observed in the data.

Intensive numerical modelling was then performed in order to reproduce the experimental data. Quantum calculations based on DFT with different functionals, different steps for probing the potential, up to 0.5 Å (is this really still reasonable, half of the N-H distance ?) and classical modelling with TRPMD. The authors eventually converged on the neural network QMD based TRPMD to best reproduce the data. They conclude that anharmonicity and nuclear quantum effect have to be accounted for to understand the low frequency vibrational features. If anharmonicity clearly appears in the experimental data and is well known to be poorly reproduced in first-principles vibrational dynamics modelling, the effect of nuclear quantum effects remains more obscure to me. NH₃ is indeed a quantum rotor showing nuclear delocalization at low temperature (see for example work by P. Schiebel or M. Prager in the late 90s). However, the 32-generated positions of the Fig. 4 describe more a disordered structure than a quantum rotor, since the protons do not look more delocalised than the nitrogen atom. Moreover the potential energy barrier of 167 meV does not favor a large tunneling splitting and coupling with librational modes. I therefore express doubts about the assignment to nuclear quantum effect based on the NNQMD-TRPMD modelling.

In conclusion, although the experimental data are of very good quality and numerical modelling has been pushed in depth to understand the vibrational features, I believe this paper is too technical to be published in Nature communications.

Reviewer #3 (Remarks to the Author):

While the authors have taken some nice data, the work is rather routine, and I think the manuscript would be better served by being published in a more specialized journal. The authors make many claims of novelty and spend much time describing theoretical methods and results that are unsurprising, as if they neglect advances over the last 15 years of research. Overall, the manuscript is fine, and the data useful, but I do not think Nature Communications is an appropriate home for this work.

The authors claim that "Compared to Raman or infrared spectroscopy, INS has no selection rules, is sensitive to hydrogen in molecular systems, and has its unique strength in discerning low to intermediate frequency modes – these are the modes that are particularly sensitive to intermolecular interactions."

This is broadly overstated, and in some cases, inaccurate. Infrared and Raman spectroscopies are sensitive to hydrogen dynamics, maybe the authors are mistakenly referring to neutron diffraction, which is sensitive to hydrogens, unlike X-ray diffraction. Moreover, INS is not unique in its ability to probe low-frequency phonons, as there are now many established methods, along with turn-key instrumentation, for acquiring THz spectra (both IR and Raman.)

*The authors state that there are challenges to accessing the low-frequency region, and cite a reference that is over a decade old. This is inaccurate given the technological advances over the last decade and availability of turn-key instruments for probing THz dynamics.

Response to Reviewer Comments

Reviewer 1

We thank the reviewer for a critical reading of the manuscript and asking important questions and clarifications. Our manuscript is improved after we have included the responses to comments and clarifications by the reviewer. Below, our response to the reviewer's comments are shown in blue font.

Comment 0: My main criticism about this study concerns the absence of high-resolution optical (Raman/IR) spectroscopic data on solid and liquid ammonia (e.g., Binbrek and Anderson, 1972, DOI:[https://doi.org/10.1016/0009-2614\(72\)80205-7](https://doi.org/10.1016/0009-2614(72)80205-7)) which, on the contrary, are very useful, especially for understanding the internal vibrations where the dispersion in Q does not play a fundamental role. The authors should include this information in their discussion, typically comparing INS and optical frequencies. This is particularly important since, exploiting these data, Zeng and Anderson in 1990 set up a complete (although approximate) force-field model of the vibrational dynamics in solid NH₃ and ND₃ (DOI: <https://doi.org/10.1002/pssb.2221620107>).

Response:

The reviewer is indeed correct in pointing out the need for comparison to optical spectroscopy data which will enhance the overall quality of the manuscript as comparisons between INS, optical spectra, and phonon calculations from atomistic dynamics can provide further insights into electronic and vibrational structure of the molecular system since they provide complementary information. For the inter-molecular modes in solid phase, we find good qualitative agreement of peak positions and major features of the density of states between optical and neutron measurements, in particular the large gap between the translational (below 23 meV) and rotational (above 28 meV) bands, which we demonstrate is heavily influenced by the zero-point motion of atoms in crystal.

For the high energy intramolecular modes, we found that the high energy N-H stretching mode as measured is harder than what is reported in optical spectroscopy. We can attribute this primarily to the recoil effect which plays a significant role at large incident energies in weakly bound molecular crystals [Tomkinson, J. The effect of recoil on the inelastic neutron scattering spectra of molecular vibrations. Chem. Phys. 127, 445-449 (1988)], and once corrected for better matches the optical data.

We have updated the introduction and discussion to reflect the above statements and added a table comparing the reported optical data and the measurements here.

Change to introduction (page 4 paragraph 1):

In addition, the calculation of INS is rigorous and straightforward if the dynamics of nuclei can be solved, and explicit treatment of electronic structure is not required. **Thus comparisons between INS, optical spectra, and phonon calculations from atomistic dynamics can provide further insights into electronic and vibrational structure of the molecular system.**

Change to main text (page 9 paragraph 2):

To get a quantitative information on the hardening of the N-H(D) stretching peaks as a result of melting, we performed double Gaussian (for ND₃) and single Gaussian (for NH₃, due to non-resolved stretching modes) fits of the neutron data with the addition of linear and Gaussian background terms which are illustrated in Figs. 2c and 2f for ND₃ and NH₃ respectively. The Gaussian background term is centered at about 450 and 325 meV for NH₃ and ND₃ respectively, and has very large full width at half maximum (FWHM≈73 and 50 meV, for NH₃ and ND₃ respectively), therefore the origin of this peak can be explained by multiphonon neutron scattering involving N-H(D) stretching and low energy intermolecular modes. For ND₃ we have compared the spectra at 5 K and just before and after the melting (at 180 and 200 K), when the phonon/vibrational populations are very similar (also, the multiphonon neutron scattering should be similar). For ND₃ we found an increase in $\omega_1 = 293 \rightarrow 297 \rightarrow 301$ meV and $\omega_2 = 312 \rightarrow 313 \rightarrow 316$ meV when transitioning from $T = 5 \rightarrow 180 \rightarrow 200$ K for the symmetric and antisymmetric peaks. For NH₃, little change was seen in the peak energy in solid phase upon further populating the phonons with increasing temperature but the shift in solid to liquid phase was clear with $\omega_{1\&2} = 417 \rightarrow 418 \rightarrow 421$ meV for $T = 5 \rightarrow 180 \rightarrow 200$ K. It is known that a recoil neutron scattering plays a significant role at large E_i in weakly bound molecular crystals³⁰ Figure S2 shows that the neutron recoil scattering on ammonia molecule was really observed in the INS spectra measured with $E_i=700$ and 500 meV, which results in shift of the observed N-H(D) stretching modes (averaged over $Q = 6 - 10 \text{ \AA}^{-1}$) to larger energy by about 8.5 meV (7.2 meV for ND₃), compared to values of the stretching modes at zero momentum transfer (the values of the stretching modes corrected on the neutron recoil are presented in the Tables S2 and S3). At incident neutron energies $E_i=280$ meV and below, the neutron recoil scattering on ammonia is insignificant (see Fig. S3).

References:

30. Tomkinson, J. The effect of recoil on the inelastic neutron scattering spectra of molecular vibrations. *Chem. Phys.* **127**, 445–449 (1988).

Change to Discussion (page 19 paragraph 2):

In comparisons with optical spectroscopy methods, we find in general good agreement with the INS data reported here^{2,49,50}. For the inter-molecular modes in solid phase, we find good qualitative agreement with peak positions and major features of the density of states, in particular the large gap between the translational and rotational bands at 32 meV, which we demonstrate is heavily influenced by the zero-point motion of atoms in crystal. For the intra-molecular spectrum in both solid and liquid phase we find the neutron spectrum for the high energy stretching modes after correction for the neutron recoil are also in agreement with optical measurements. A table comparing these results is provided in supplementary tables S2 and S3.

References:

2. Binbrek, O. S. & Anderson, A. Raman spectra of molecular crystals. Ammonia and 3-deutero-ammonia. *Chem. Phys. Lett.* **15**, 421–427 (1972).
49. Ujike, T. & Tominaga, Y. Raman spectral analysis of liquid ammonia and aqueous solution of ammonia. *J. Raman Spectrosc.* **33**, 485–493 (2002).
50. Zeng, W. Y. & Anderson, A. Lattice Dynamics of Ammonia. *Phys. status solidi* **162**, 111–

Changes to supplemental information:

Comparison of Optical and Neutron Data:

Below table S2 compares optical data taken in main text references 2,48,49 for the solid phase to those measured by VISION and SEQUOIA for the inter-molecular modes and those taken by SEQUOIA for the intra-molecular modes for NH₃. A similar comparison in the liquid phase for the inter-molecular modes is given table S3.

	Optical 18K	Neutron 5K (SEQUOIA, VISION)
Acoustic	--	8.9, 8.75
Acoustic	--	10.9,10.65
Γ Translational	13.39	12.7, 12.4
M Translational	--	14.7,14.5
M Translational	--	16.0,15.5
Γ Translational	17.48	19.0,18.6
Γ Translational	17.48	19.0,18.6
Γ Translational	22.8	21.2,20.8
Rotational	--	30.4,29.75
Rotational	32.4	33.0,32.3
Rotational	37.13	38.4,39.5
Rotational	38.8	39.8,39.5
Rotational	44.72 & 45.47	44.5, 44.3
Rotational	--	55.2,54.0
Rotational	--	60.5,59.75
Rotational	66.08	67
Symm. Bend	131.05	133
Symm. Bend	132.97	133
Degen. Bend	202.83	203
Degen. Bend	204.57	207
Degen. Bend	208.04	
Symm. Stretch	397.18	Non-resolved 408.5
Degen. Anti-Symm. Stretch	417.82	Non-resolved 408.5
Degen. Anti-Symm. Stretch	418.8	

Table S2. Optical – neutron data comparison in solid phase.

	Optical (200K)	Neutron (200K)
Symm. Bend	132	132
Degen. Bend	203	205
Symm. Stretch	398	Non-resolved 412.5
Degen. Anti-Symm. Stretch	409	Non-resolved 412.5
Degen. Anti-Symm. Stretch	419	

Table S3. Optical – neutron data comparison in liquid phase.

Comment 1: The experimental INS data on solid ammonia at 106 K reported by Goyal et al. (1972) should be explicitly cited (DOI: <https://doi.org/10.1002/pssb.2220500232>).

Response

We have explicitly cited this reference and discussed its importance in the introduction and discussion.

Change to introduction (page 4, paragraph 2):

However, to the best of our knowledge, there is no available INS data for the phonon density states covering the entire vibrational spectrum along the solid to liquid phase transition for ammonia of high enough quality to be rigorously compared to theoretical models. The only INS available data are from Goyal et al. in 1972¹⁵, which only includes one temperature in solid phase, and the work of Jack Carpenter *et al.* from 2004¹⁶, which does not have sufficient resolution, especially in the energy range of inter-molecular interactions, to be rigorously compared to physical models for ammonia in its liquid and solid phases.

References:

- Goyal, P. S., Dasannacharya, B. A., Thaper, C. L. & Iyengar, P. K. Frequency distribution function of solid ammonia. *Phys. status solidi* 50, 701–708 (1972).
- Carpenter, J., Micklich, B. & Zanotti, J. M. Neutron scattering measurements from cryogenic ammonia: a progress report. in *ACoM-6 - 6th international workshop on advanced cold moderators Proceedings* 236 (2004).

Change to discussion (page 18, paragraph 1):

We have performed INS measurements on solid and liquid ammonia and compared the measurements to DFT simulations. We find NQE induced anharmonicity that fundamentally changes the predicted spectrum with conventional DFT simulations, which we illustrate through neural network-based PIMD simulations using the TRPMD implementation of PIMD. PIMD

simulations can reproduce the hardening of N-H stretching modes in the liquid phase. The hardening was determined to be due to different spatial and temporal characters of the hydrogen bonds. In solid phase, the constant and percolated hydrogen bonding network makes the N-H stretching modes softer than in the liquid phase, where brief periods of strong inter-molecular interaction are followed by periods of low/non-interaction. **The reported solid phase INS measurements on density of states is in good agreement with that by Goyal et al. in 1972, but with much enhanced resolution¹⁵.**

References :

15. Goyal, P. S., Dasannacharya, B. A., Thaper, C. L. & Iyengar, P. K. Frequency distribution function of solid ammonia. *Phys. status solidi* **50**, 701–708 (1972).

Comment 2: Page 5, lines 1-7: Dealing with quantum effects and anharmonic effects in the ammonia microscopic dynamics there are some statements which sound slightly vague and hand-waving. I think that this theoretical explanation should be surely short but also rigorous: pseudo-harmonic lattice dynamics is exact from a quantum point of view, but it needs that the interparticle potential can be well approximated by a harmonic term. Since it is not often the case, one speaks of “anharmonic effects”, which are relevant when the particle mean square displacement is large (i.e., at high temperature or for low particle masses). On the contrary, Newtonian molecular dynamics deals with anharmonic potentials very easily, but it is unable to include “quantum effects” since it is purely classical. Quantum effects are relevant when the particle zero-point energy is not negligible with respect to the average thermal energy (i.e., at low temperature or for low particle masses) This is exactly the reason why one switches to RPMD which can deal with both effects, provided that quantum effects are not too strong.

Response:

The reviewer is correct for pointing out need to clarify these subtle points to enhance the clarity of the manuscript. By nuclear quantum effect we are referring to the zero-point energy. At 5 K the discrepancy between the measured and computed INS spectra due to “anharmonic” effect would not come from thermal motion which can be handled by classical molecular dynamics but by extension of the atomic phase space to incorporate the zero-point motion of the nuclei. We have added the following changes to manuscript to address this.

Change to page 12 paragraph 1:

It points two possibilities: 1) any available model cannot describe the true van der Waals interaction; or 2) these modes are highly anharmonic thus, the harmonic approximation as used by DFPT failed. Since the discrepancy is seen with the data obtained at 5 K, **the anharmonicity is caused by extension of the atomic phase space due to zero-point motion of the atoms rather than their thermal motion.**

Change to page 13 paragraph 3 and page 14 paragraph 2:

The anharmonicity of a particular mode (in this case, the NH₃ libration) can also be evaluated by mapping out the potential energy profile corresponding to the mode, which can be obtained by nudged elastic band calculations. The result in Fig. 4b shows that the three-fold potential energy profile has a barrier of about 167 meV. As a quantum rotor in this potential well,

the excitation energies can be predicted using a quantum rotor model implemented in DAVE⁴⁶, which are illustrated in supplemental table S1. The first two energies ($\sim 1.3 \times 10^{-6}$ meV) are the tunneling splitting of the rotational mode which is beyond our instrument resolution. The next three energies (32 meV) correspond to the excitation of the rotational vibration of NH₃ (corresponding to $n=0 \rightarrow 1$ excitation in a quantum oscillator model). This energy represents what is directly measured in our INS experiment and is in good agreement with the VISION measurement for this mode.

The above analysis highlights the problem but has yet to offer a general solution, as the quantum rotor model cannot be used to simulate the entire INS spectra or easily generalized to study other modes/systems. While conventional molecular dynamics can accurately describe anharmonic effects due to finite temperature, it cannot capture anharmonic shifts due to the zero-point motion of the atoms which is what captured by the quantum rotor model $n=0 \rightarrow 1$ excitation. A promising solution is path integral molecular dynamics (PIMD), in which the quantum partition function is mapped to a classical analogue by using replicas (beads) connected by springs (ring polymers) to represent each atom¹⁹, which effectively extends the phase space of the atomic system due to their zero point motion. This method is suited for problems when the particle zero-point energy is not negligible with respect to the average thermal energy, which is expected for hydrogen systems at low temperature such as ammonia.

References:

19. Feynman, R. P. & Hibbs, A. R. Quantum Mechanics and Path Integrals. McGraw-Hill, New-York. (1965).
46. Azuah, R. T. et al. DAVE: a comprehensive software suite for the reduction, visualization, and analysis of low energy neutron spectroscopic data. J. Res. Natl. Inst. Stand. Technol. 114, 341 (2009).

Change to SI

Excitation Energies (meV)
1.30×10^{-6}
1.30×10^{-6}
32.14
32.14
32.14

Table S1: First 5 energies in quantum rotor model. The first two energies correspond to the tunnel splitting, while the next three are to $n=0 \rightarrow 1$ excitation in a quantum oscillator model representing the excitation of the rotational vibration of NH₃. The latter three are what are measured in an INS experiment.

Comment 3: Page 7, Eq. (1) and following lines: The use of the Q integration to get rid of the coherent components in the ND_3 INS spectra in order to establish a comparison with the INS spectra from NH_3 is a very common practice and is based on the so-called “Bredov

approximation” (see, e.g., the study by N. Breuer, 1974, DOI: <https://doi.org/10.1007/BF01677937>) which, however, is rigorously proven only in the case of simple Bravais systems. I think that the authors should briefly mention this issue.

Response:

Yes, it is well known that the correct (or generalized) density of phonon (or vibrational) states for coherently scattering sample can be obtained from INS spectra by summation over Q-range, which is much larger than the Brillouin zone of the crystal, to have a good averaging of the spectra.

We have added the following change to the manuscript (page 7 paragraph 2):

Fig. 1, a and b, shows integrated dynamic structure factor:

$$S(E) = \int dQ S(Q, E) \quad (1)$$

measured with the SEQUOIA spectrometer for deuterated ammonia (ND₃) with incident energies in the range for intermolecular vibrations of ND₃, which represents the phonon density of states. The INS spectra for protonated ammonia (NH₃) measured with the VISION spectrometer at base temperature 5 K, and 60 K are illustrated in Fig. 1c. The integrated dynamic structure from SEQUOIA for protonated ammonia is shown in supplemental Fig. S1, which is in good agreement with the measurements taken with VISION. The coherent (incoherent) neutron scattering cross sections for H, D and N atoms are 1.76 b (80.26 b), 5.59 b (2.05 b) and 11.01 b (0.50 b), respectively. For accurate determinations of the phonon density of states when the scattering is predominantly coherent, as in the case of ND₃, it is very important to average the measured neutron scattering over a large volume of reciprocal space for the resulting data to reflect the true phonon density of states²⁹. In the current experiment the ratio of the momentum transfer coverage to the Brillouin zone volume was about 20 for the lowest incident energy of 30 meV (and this value is larger for larger incident energy), therefore the condition of averaging was valid in the whole range of energy transfer studied.

Fig. 1, a and b, demonstrates that with increasing temperature past 100 K towards the melting point (195 K), peaks in the acoustic and optical regimes display a strong anharmonic softening.

References :

29. Breuer, N. Determination of the phonon spectrum from coherent neutron scattering by polycrystals. *Zeitschrift für Phys.* **271**, 289–293 (1974).

Comment 4: Pages 8 and 9: In connection with reconstruction of the INS spectra (starting from the velocity autocorrelation functions calculated via RPMD), where both fundamental and multiphonon contributions are present, the authors should clearly state that their procedure (embedded, for example, in the formulas used by the OCLIMAX code) is exact only in an essentially harmonic context. If applied to anharmonic systems (including, e.g., liquids) this approach is well-known under the name of “Gaussian approximation” (see Rahman et al., 1962 DOI: <https://doi.org/10.1103/PhysRev.126.986>). Especially in the case of liquid ammonia at 200 K, additional care is needed to implement such a multiphonon calculation, since, rigorously

speaking, the standard Debye-Waller coefficient does not exist any longer and the self diffusion coefficient causes a noticeable peak broadening, especially at large Q values.

Response:

We agree that the multi-phonon calculations in OCLIMAX are made under harmonic approximation, which do not account for shift of the multi-phonon peaks due to anharmonicity. Nevertheless, the total spectra calculated with OCLIMAX provide a reasonable agreement with the experimental spectra (mostly due to including of 10 orders of multi-phonon excitations, which results in almost non-structured multi-phonon background).

Change to page 17 paragraph 2:

Fig. 5g shows the computed INS spectrum including multi-phonon scattering for solid and liquid NH₃, where we also see a hardening in high energy N-H stretching modes compared to those computed in the crystalline phase, which indicates weaker inter-molecular interactions in the liquid. While care must be taken when incorporating the multi-phonon shifts within the harmonic approximation in the so called “Gaussian approximation”⁴⁸, especially in the liquid phase. In OCLIMAX we used a direct convolution of the low- and high-energy modes (up to 10 orders, but still in harmonic approximation) and as a result the overall agreement is still reasonable with experiment. The good agreement in liquid is largely due to the phonon structure in liquid phase at low energy both computed and measured is highly unstructured resulting unstructured multi-phonon background.

References:

48. Rahman, A., Singwi, K. S. & Sjölander, A. Theory of Slow Neutron Scattering by Liquids. *I. Phys. Rev.* **126**, 986–996 (1962).

Response to Reviewer Comments

Reviewer 2

We thank the reviewer for a critical reading of the manuscript and asking important questions and clarifications. Our manuscript is improved after we have included the responses to comments and clarifications by the reviewer. Below, our response to the reviewer's comments are shown in blue font.

Comment 1: Two spectrometers were used, SEQUOIA and VISION in the Oak Ridge neutron center (SNS), although it is not clear what the information obtained on VISION brings after the measurements performed on SEQUOIA.

Response:

In addition to SEQUOIA spectrometer, we also used VISION spectrometer because it provides better statistics and energy resolution at energies below 100 meV.

Comment 2: Quantum calculations based on DFT with different functionals, different steps for probing the potential, up to 0.5 Å (is this really still reasonable, half of the N-H distance ?) and classical modelling with TRPMD.

Response:

The finite displacement simulation was meant to roughly probe the effects of anharmonicity on different vibrational modes, and to reveal that the librational modes have a much stronger anharmonicity than the translational modes. The displacement (0.1 to 0.5 Å) was chosen to cover a sufficiently wide range to reveal these effects, but they may not represent the actual atomic displacement or how the atoms move in reality (that is why none of the spectra simulated by FDM matches experiment in the full energy range, and we need to perform TRPMD for quantitative agreement). We agree that the maximum displacement used (0.5 Å) has exceeded what has been measured experimentally (even considering that the MSD in NH₃ is larger than that in ND₃), and we have added a note in the manuscript to avoid misunderstanding.

Change to page 13 paragraph 2:

This is a clear indication that the librational modes are anharmonic, and the corresponding potential energy profile is non-parabolic, **but the displacement method is not meant to be quantitatively accurate with 0.5 Å being much larger than the expected mean-square displacement of the atoms at 5 K and accordingly the other parts of the spectra do not show a consistent improvement.**

Comment 3: They conclude that anharmonicity and nuclear quantum effect have to be accounted for to understand the low frequency vibrational features. If anharmonicity clearly appears in the experimental data and is well known to be poorly reproduced in first-principles vibrational dynamics modelling, the effect of nuclear quantum effects remains more obscure to me. NH₃ is indeed a quantum rotor showing nuclear delocalization at low temperature (see for example work

by P. Schiebel or M. Prager in the late 90s). However, the 32-generated positions of the Fig. 4 describe more a disordered structure than a quantum rotor, since the protons do not look more delocalised than the nitrogen atom. Moreover the potential energy barrier of 167 meV does not favor a large tunneling splitting and coupling with librational modes. I therefore express doubts about the assignment to nuclear quantum effect based on the NNQMD-TRPMD modelling.

Response:

With regards to nuclear quantum effect, we are referring to the extension of atomic phase space due to the zero-point motion of the atomic nuclei which is what is represented by the 32 positions generated by TRPMD. At 5 K, discrepancies in the computed and measured spectra from the anharmonic nature of the ammonia lattice would be due to this extended phase space from the zero-point motion as there is little thermal energy available. The quantum rotor model was used to directly compute the zero-point energy of NH₃ rotational mode to provide justification for the TRPMD to be a valid approach to assess the discrepancies of those measured by experiment and those computed using diagonalizing the dynamical matrix computed using density functional theory. The reviewer is correct in pointing that computed energy barrier of 167 meV is not conducive to tunneling.

The quantum rotational transitions of NH₃ that we solve using the hindered rotor mode is implemented in the DAVE software. Using the rotational constant of 0.768 meV for NH₃, and an energy barrier of 167 meV, we solve the excitation energies as listed in the right column of figure R1. The first two energies (1.3×10^{-6} meV) are the tunneling splitting of the ground state which are indeed very small (and beyond our instrument resolution). The next three energies (32 meV) are due to the rotational vibration of the NH₃ (corresponding to n= 0→1 excitation in a quantum oscillator model). This is what we measured with INS and what is simulated with TRPMD.

Figure R1. DAVE calculation of quantum rotor model.

With regards to localization of nitrogen atom versus the hydrogen atom, figure 4C is a visualization of beads for atoms composing a single ammonia molecule taken from an *ab-initio* PIMD trajectory at 60 K to illustrate the difference in the PIMD and standard MD method. Each of the beads are fictitious particles within which ensemble averages are taken over to computer observable quantities such as the INS spectrum.

As we increase the number of frames visualized, the beads effectively represent the phase space visited by the nitrogen and hydrogen atoms due to both thermal and zero-point motion. Figure R2 below shows the bead positions for one ammonia molecule for 1,000 frames, illustrating the nitrogen atom is far more localized than the hydrogen atoms.

Figure R2. Beads of Ammonia Molecule in PIMD trajectory. Nitrogen colored in blue and hydrogen in white. Nitrogen and Hydrogen spheres are drawn to same size to better visualize effective phase space visited by both species.

Change to page 12 paragraph 2.

It points two possibilities: 1) any available model cannot describe the true van der Waals interaction; or 2) these modes are highly anharmonic thus, the harmonic approximation as used by DFPT failed. Since the discrepancy is seen with the data obtained at 5 K, **the anharmonicity is caused by extension of the atomic phase space due to zero-point motion of the atoms rather than their thermal motion.**

Change to page 13 paragraph 3 and page 14 paragraph 2

The anharmonicity of a particular mode (in this case, the NH_3 libration) can also be evaluated by mapping out the potential energy profile corresponding to the mode, which can be obtained by nudged elastic band calculations. The result in Fig. 4b shows that the three-fold

potential energy profile has a barrier of about 167 meV. As a quantum rotor in this potential well, the excitation energies can be predicted using a quantum rotor model implemented in DAVE⁴⁶, which are illustrated in supplemental table S1. The first two energies ($\sim 1.3 \times 10^{-6}$ meV) are the tunneling splitting of the rotational mode which is beyond our instrument resolution. The next three energies (32 meV) correspond to the excitation of the rotational vibration of NH₃ (corresponding to $n=0 \rightarrow 1$ excitation in a quantum oscillator model). This energy represents what is directly measured in our INS experiment and is in good agreement with the VISION measurement for this mode.

The above analysis highlights the problem but has yet to offer a general solution, as the quantum rotor model cannot be used to simulate the entire INS spectra or easily generalized to study other modes/systems. While conventional molecular dynamics can accurately describe anharmonic effects due to finite temperature, it cannot capture anharmonic shifts due to the zero-point motion of the atoms which is what captured by the quantum rotor model $n=0 \rightarrow 1$ excitation. A promising solution is path integral molecular dynamics (PIMD), in which the quantum partition function is mapped to a classical analogue by using replicas (beads) connected by springs (ring polymers) to represent each atom¹⁹, which effectively extends the phase space of the atomic system due to their zero point motion. This method is suited for problems when the particle zero-point energy is not negligible with respect to the average thermal energy, which is expected for hydrogen systems at low temperature such as ammonia. The background of Fig. 4c shows a typical first principles-based simulation, where the atoms are treated classically, and the electron charge density is treated quantum-mechanically to compute atomic forces, illustrated as gray iso-surfaces. In the foreground, we have highlighted one NH₃ molecule from a PIMD simulation of the same atomic configuration, where each atom has 32 replicas that are harmonically coupled together. As we increase the number of frames visualized, the beads effectively represent the phase space visited by the nitrogen and hydrogen atoms due to both their thermal and zero-point motion. Figure S4 shows bead positions for one ammonia molecule for 1000 frames in an *ab-initio* PIMD simulation illustrating the large phase space visited by the light hydrogen atoms.

References :

19. Feynman, R. P. & Hibbs, A. R. Quantum Mechanics and Path Integrals. McGraw-Hill, New-York. (1965).
46. Azuah, R. T. et al. DAVE: a comprehensive software suite for the reduction, visualization, and analysis of low energy neutron spectroscopic data. J. Res. Natl. Inst. Stand. Technol. 114, 341 (2009).

Change to SI

Table S1: First 5 energies in quantum rotor model. The first two energies correspond to the tunnel splitting, while the next three are to $n=0 \rightarrow 1$ excitation in a quantum oscillator model representing the excitation of the rotational vibration of NH₃. The latter three are what are measured in an INS experiment.

Excitation Energies (meV)

1.30×10^{-6}
1.30×10^{-6}
32.14
32.14
32.14

1000 frames

1 frame

Figure S4. Beads of Ammonia Molecule in PIMD trajectory. Nitrogen colored in blue and hydrogen in white. Nitrogen and Hydrogen spheres are drawn to same size to better visualize effective phase space visited by both species.

Response to Reviewer Comments

Reviewer 3

We thank the reviewer for a critical reading of the manuscript and asking important questions and clarifications. Our manuscript is improved after we have included the responses to comments and clarifications by the reviewer. Below, our response to the reviewer's comments are shown in blue font.

Comment: The authors claim that Infrared and Raman spectroscopies are sensitive to hydrogen dynamics, maybe the authors are mistakenly referring to neutron diffraction, which is sensitive to hydrogens, unlike X-ray diffraction. Moreover, INS is not unique in its ability to probe low-frequency phonons, as there are now many established methods, along with turn-key instrumentation, for acquiring THz spectra (both IR and Raman.)

*The authors state that there are challenges to accessing the low-frequency region, and cite a reference that is over a decade old. This is inaccurate given the technological advances over the last decade and availability of turn-key instruments for probing THz dynamics.

Response

We agree that modern THz spectroscopy is very powerful method to study low-frequency phonons, but it is still limited to very low momentum transfer. It can cover only up to $\sim 3 \times 10^{-4}$ Å, which is almost gamma-point in the Brillouin zone of the typical crystals. As such, THz spectroscopy is useful to get information on the vibrational modes around the gamma point, then we need to do extra calculations to construct the total dispersion curves and density of phonon states. On the contrary, this information can be obtained directly from the INS measurements of the coherently scattering samples like ND₃.

In addition, comparisons between INS, optical spectra, and phonon calculations from atomistic dynamics can provide further insights into electronic and vibrational structure of the molecular system as they do not explicitly measure the same thing. Our work provides INS data for the entire solid-liquid phase transition, which is currently lacking, along with rigorous and robust theoretical analysis framework, which will have applications beyond ammonia.

To highlight the role of this work within the boarder context of vibrational spectroscopy we have added a discussion comparing our results to various optical methods without diminishing the importance of these techniques.

Change to introduction (page 4 paragraph 1).

Despite the importance, quantitative understanding of molecular vibrations is only sometimes straightforward. Experimentally, widely used spectroscopy methods such as Raman and Infrared scattering are only able to measure a subset of the vibrational modes due to the lack of momentum from photons, the selection rule **and has historically been challenging** to probe the low frequency region¹⁴ (sub-THz). **While modern terahertz spectroscopy has become a very**

powerful tool to study low-frequency phonons, it is still limited to very low momentum transfer (Q) and can cover only up to $\sim 3 \times 10^{-4} \text{ \AA}^{-1}$, which is essentially only the gamma-point in the Brillion zone of the typical crystals and careful analysis is needed to reconstruct the dispersive properties and density of states. Inelastic neutron scattering (INS) is ideally suited to measure the full phonon density states and for strongly coherent scatters the Q dependence of phonons can easily be obtained. In addition, the calculation of INS is rigorous and straightforward if the dynamics of nuclei can be solved, and explicit treatment of electronic structure is not required. Thus, comparisons between INS, optical spectra, and phonon calculations from atomistic dynamics can provide further insights into electronic and vibrational structure of the molecular system.

Change to Discussion (page 19 paragraph 2):

In comparisons with optical spectroscopy methods, we find in general good agreement with the INS data reported here^{2,49,50}. For the inter-molecular modes in solid phase, we find good qualitative agreement with peak positions and major features of the density of states, in particular the large gap between the translational and rotational bands at 32 meV, which we demonstrate is heavily influenced by the zero-point motion of atoms in crystal. For the intra-molecular spectrum in both solid and liquid phase we find the neutron spectrum for the high energy stretching modes after correction for the neutron recoil are also in agreement with optical measurements. A table comparing these results is provided in supplementary tables S2 and S3.

References :

2. Binbrek, O. S. & Anderson, A. Raman spectra of molecular crystals. Ammonia and 3-deutero-ammonia. *Chem. Phys. Lett.* **15**, 421–427 (1972).
49. Ujike, T. & Tominaga, Y. Raman spectral analysis of liquid ammonia and aqueous solution of ammonia. *J. Raman Spectrosc.* **33**, 485–493 (2002).
50. Zeng, W. Y. & Anderson, A. Lattice Dynamics of Ammonia. *Phys. status solidi* **162**, 111–117 (1990).

Changes to Supplemental Information

Comparison of Optical and Neutron Data:

Below table S2 compares optical data taken in main text references 2,48,49 for the solid phase to those measured by VISION and SEQUOIA for the inter-molecular modes and those taken by SEQUOIA for the intra-molecular modes for NH₃. A similar comparison in the liquid phase for the inter-molecular modes is given table S3.

Table S2. Optical – neutron data comparison in solid phase.

	Optical 18K	Neutron 5K (SEQUOIA, VISION)
Acoustic	--	8.9, 8.75
Acoustic	--	10.9, 10.65

Γ Translational	13.39	12.7, 12.4
M Translational	--	14.7,14.5
M Translational	--	16.0,15.5
Γ Translational	17.48	19.0,18.6
Γ Translational	17.48	19.0,18.6
Γ Translational	22.8	21.2,20.8
Rotational	--	30.4,29.75
Rotational	32.4	33.0,32.3
Rotational	37.13	38.4,39.5
Rotational	38.8	39.8,39.5
Rotational	44.72 & 45.47	44.5, 44.3
Rotational	--	55.2,54.0
Rotational	--	60.5,59.75
Rotational	66.08	67
Symm. Bend	131.05	133
Symm. Bend	132.97	133
Degen. Bend	202.83	203
Degen. Bend	204.57	207
Degen. Bend	208.04	
Symm. Stretch	397.18	Non-resolved 408.5
Degen. Anti-Symm. Stretch	417.82	Non-resolved 408.5
Degen. Anti-Symm. Stretch	418.8	

Table S3. Optical – neutron data comparison in liquid phase.

	Optical (200K)	Neutron (200K)
Symm. Bend	132	132
Degen. Bend	203	205
Symm. Stretch	398	Non-resolved 412.5
Degen. Anti-Symm. Stretch	409	Non-resolved 412.5
Degen. Anti-Symm. Stretch	419	

REVIEWERS' COMMENTS

Reviewer #1 (Remarks to the Author):

The manuscript NCOMMS-23-38777A, entitled "Neutron Scattering, Neutron Quantum Effects and Neural Networks: The Delicate Case of the Ammonia Vibrational spectroscopy" by T. M. Linker et al., resubmitted for publication on "Nature Communications", has been newly reviewed.

The authors have substantially improved the quality of their paper and have fully replied to the reviewers' comments in a satisfactory way.

As I have written in my first review, this paper is original, scientifically sound, globally clear, and professionally written.

For these reasons I surely recommend its publications in the present revised form.

Reviewer #2 (Remarks to the Author):

Although the comments have been addressed, I still find the paper quite technical for the journal.

Reviewer #3 (Remarks to the Author):

The authors have attempted to respond to the previous referees, and have done an adequate job of doing so. Unfortunately, my opinion on the novelty and impact of the work remains unchanged - I do not see why this work belongs in Nature Communications, and it is much more appropriate for a more specialized journal. This is something that I would expect to see in a 'standard' physical chemistry journal, such as J. Phys. Chem., and I am not sure what makes this work noteworthy-enough to be published in the general Nature Communications journal. It is technically good work, but rather routine, and ultimately the results are not that spectacular.

And, for what its worth, changing the phrasing on the terahertz comments to simply add 'and has historically been challenging', but not adding any new references or anything further is a huge cop-out and it is offensive to this Referee that the authors fail to acknowledge (1) the original comment from the Reviewer, and (2) the huge body of work that has emerged since the publication of the referenced paper. Furthermore, IR spectroscopy is not scattering, its absorption. This article is sloppy and full of similar mistakes.

If the editors disagree with my assessment, then by all means, publish this in Nature Communications.

Response to Reviewer Comments :

Response to Reviewer 3

Comment :

The authors have attempted to respond to the previous referees, and have done an adequate job of doing so. Unfortunately, my opinion on the novelty and impact of the work remains unchanged -- I do not see why this work belongs in Nature Communications, and it is much more appropriate for a more specialized journal. This is something that I would expect to see in a 'standard' physical chemistry journal, such as J. Phys. Chem., and I am not sure what makes this work noteworthy-enough to be published in the general Nature Communications journal. It is technically good work, but rather routine, and ultimately the results are not that spectacular.

And, for what its worth, changing the phrasing on the terahertz comments to simply add 'and has historically been challenging', but not adding any new references or anything further is a huge cop-out and it is offensive to this Referee that the authors fail to acknowledge (1) the original comment from the Reviewer, and (2) the huge body of work that has emerged since the publication of the referenced paper. Furthermore, IR spectroscopy is not scattering, its absorption. This article is sloppy and full of similar mistakes.

If the editors disagree with my assessment, then by all means, publish this in Nature Communications.

Response :

To address the remaining concerns of reviewer #3 we have removed statements that can be subjectively interpreted regarding previous vibrational spectroscopy studies, while still highlighting the novelty of the work within the broader context of vibrational spectroscopy.

In the introduction we have removed comments on the difficulties of THz spectroscopy and focused on the advantages of neutron spectroscopy as a tool and have explained why obtaining high resolution phonon density of states covering the entire vibrational spectrum along the solid to liquid phase transition for ammonia is important/useful. In addition, we have modified the intro to review what has been studied until now removing emphasis on the limitations of past studies except when necessary to highlight the strengths of what is now possible with modern neutron facilities and advanced computational techniques that incorporate machine learning in comparison to what data and theoretical analysis was performed in the past for ammonia.

Change to the Introduction :

For measuring the full vibrational density of states, inelastic neutron scattering is a powerful tool that can easily access low frequency regions (sub-THz) and for strongly coherent scatterers the Q dependence of the phonons can straightforwardly be obtained¹⁴. In addition, the calculation of INS is rigorous and straightforward if the dynamics of nuclei can be solved, and

explicit treatment of electronic structure is not required. Thus, comparisons between INS, optical spectra, and phonon calculations from atomistic dynamics can provide further insights into electronic and vibrational structure of the molecular system. All these features make INS an appealing technique for studying phonons in molecular solids and liquids.

High-quality INS data is necessary to develop accurate models of the dynamic behavior of molecular solids and liquids, as multiple complicating factors require careful considerations, such as van der Waals interactions, nuclear quantum effects (NQEs), and phonon anharmonicity. Goyal et al. in 1972¹⁵ was able to investigate the inter-molecular spectrum at one temperature in solid phase for ammonia, and Jack Carpenter *et al.* from 2004¹⁶ measured the density of states at both solid and liquid phases up to 250 meV, but lacks enough energy points for rigorous comparison theoretical models of the fine vibrational structure (especially at inter-molecular energies), and the dispersion (Q dependence) was not measured nor the high energy stretching modes. With modern neutron facilities and advanced simulation techniques, it is now possible to obtain high resolution neutron data along the full range of vibrational energies at multiple temperatures in solid and liquid phase and compare these results to different physical models that can consider van der Waals interactions, NQEs, and phonon anharmonicity on a first principles basis.

In particular understanding the role of van der Waals force is particularly relevant in molecular systems as it is a significant part, if not a dominant part, of the intermolecular interactions⁶. The conventional density functional theory (DFT) cannot describe van der Waals interactions, and empirical corrections are often included, leaving additional uncertainties when modeling such systems. Moreover, most molecular solids contain light elements such as H, for which NQEs could be significant, especially at low temperatures (even though the impact can

also be observed at room temperature). Conventional lattice dynamics or molecular dynamics treat nuclei as classical point particles with no spread; thus NQEs are not considered. Last but not least, molecular solids are usually “soft” and tend to exhibit phonon anharmonicity. Such anharmonicity could be coupled with NQEs, making the analysis even more demanding.

Traditionally incorporating all the described effects into one physical model is extremely challenging due to the excessive computational cost. For example, *ab initio* path integral molecular dynamics (PIMD) simulations^{19–22} based on DFT allows one to consider NQEs, phonon anharmonicity, and van der Waals interactions (within the chosen DFT exchange-correlation functional); however, it is extremely costly as it requires multiple replica DFT simulations to be performed. As most of the computational expense for *ab initio* PIMD simulations comes from having to compute multiple replica DFT simulations, the computational cost can be greatly decreased if the underlying DFT simulations can be replaced by much cheaper computational models.

In this regard, neural-network quantum molecular dynamics (NNQMD) simulations²³ based on machine learning offer a promising tool reduce the computational cost as they revolutionize atomistic modeling of materials by following the trajectories of all atoms with quantum-mechanical accuracy at a drastically reduced computational cost. NNQMD can not only predict accurate interatomic forces but can capture quantum properties such as electronic polarization²⁴ and electronic excitation²⁵, thus the ‘Q’ in NNQMD. A more recent breakthrough in NNQMD has drastically improved the accuracy of force prediction over those previous models, which was achieved through rotationally equivariant neural networks based on a group theoretical formulation of tensor fields^{26–28}. Thus combining PIMD simulations with NNQMD, one can obtain highly accurate first principles based prediction of the INS spectrum.

Here we report measured vibrational density of states and dynamic structure factor for deuterated and protonated ammonia along the solid-to-liquid phase transition with inelastic neutron scattering using SEQUOIA¹⁷ and VISION¹⁸ spectrometers at Oak Ridge National Laboratory, **and their comparison to DFT and NNQMD based simulations.** Our measured INS spectrum shows strongly anharmonic behavior of the inter-molecular phonon dynamics in solid phase. However, little change in the vibrational spectrum for the intra-molecular modes is observed as a function of temperature in solid phase. In the liquid phase we find hardening of the high energy N-H stretching modes compared to that of the solid phase, which indicates a decrease in the strength of inter-molecular interactions in the liquid phase. **We find standard DFT simulations are highly sensitive to the choice of the van der Waals correction to the exchange functional and fail to reproduce the INS spectrum. Through *ab-initio* PIMD and large scale NNQMD-based PIMD simulations we illustrate the discrepancy comes from phonon anharmonicity and its coupling with NQEs.** The introduced computational approach **to model the INS spectrum** is scalable to any material system, offering a robust method to quantify the role of NQEs on material vibrational dynamics.